# Achieving Faster than $O(1/t)$ Convergence in Convex Federated Learning

## Abstract

This paper aims to achieve faster than $O(1/t)$ convergence in federated learning for general convex loss functions. Under the independent and identical distribution (IID) condition, we show that accurate convergence to an optimal solution can be achieved in convex federated learning even when individual clients select stepsizes locally without any coordination. More importantly, this local stepsize strategy allows exploitation of the local geometry of individual clients' loss functions, and is shown to lead to faster convergence than the case where a same universal stepsize is used for all clients. Then, when the distribution is non-IID, we employ the sharing of gradients besides the global model parameter to ensure $o(1/t)$ convergence to an optimal solution in convex federated learning. For both algorithms, we theoretically prove that stepsizes that are much larger than existing counterparts are allowed, which leads to much faster convergence in empirical evaluations. It is worth noting that, beyond providing a general framework for federated learning with drift correction, our second algorithm's achievement of $o(1/t)$ convergence to the exact optimal solution under general convex loss functions has not been previously reported in the federated learning literature—except in certain restricted convex cases with additional constraints. We believe that this is significant because even after incorporating momentum, existing first-order federated learning algorithms can only ensure $O(1/t)$ convergence for general convex loss functions when no additional assumptions on heterogeneity are imposed.

## 1 Introduction

Federated learning has received intensive attention since it was proposed by McMahan et al. (2017). Nowadays, it has found applications in diverse areas including healthcare (Xu et al., 2021a; Nguyen et al., 2022; Antunes et al., 2022), smart cities (Pandya et al., 2023; Jiang et al., 2020; Ramu et al., 2022), natural language processing (Liu et al., 2021; Lin et al., 2021; Zhu et al., 2020), the Internet of things (Nguyen et al., 2021; Zhang et al., 2022b; Ghimire & Rawat, 2022), among others. In federated learning, the training data sets are located on individual clients which cooperatively learn a common model via periodically sharing their intermediate learning results with a central server (McMahan et al., 2017). Compared to centralized learning where all data are aggregated to a data center, federated learning has many advantages, such as enhanced security (Ma et al., 2020; Mothukuri et al., 2021; Zhang et al., 2022a), better privacy (Yang et al., 2019; Agarwal et al., 2018; Li et al., 2020), and higher communication efficiency (Sattler et al., 2019; Chen et al., 2021; Hamer et al., 2020). To date, many aspects of federated learning have been extensively studied, including stepsize design (see, e.g., Kim et al. (2023); Mukherjee et al. (2023); Pan et al. (2023)), communication efficiency (see, e.g., Nori et al. (2021); Tran et al. (2019); Liu et al. (2022)), optimization mechanism (see, e.g., Luo et al. (2021); Wei et al. (2024); Feng et al. (2021)), among others.

In federated learning, clients perform multiple local training steps before communicating with a central server to reduce the burden of information transmission (McMahan et al., 2017). However, these local training steps move local optimization variables toward the minima of local loss functions and introduce a drift from the optimal solution of the global loss function. Therefore, when the data distribution is non-IID among the clients, local training steps result in slow convergence and learning errors, which is called the "client-drift phenomenon" (Karimireddy et al., 2020; Li et al., 2019; Malinovskiy et al., 2020; Charles & Konečnỳ, 2020; Charles & Konečný, 2021; Pathak & Wainwright, 2020). In fact, under non-IID data, popular federated learning algorithms, such as

FedAvg, can only ensure accurate convergence under diminishing stepsizes, which, however, results in slow convergence Mitra et al. (2021b). It is worth noting that by imposing additional assumptions on the loss function (e.g., the interpolation and the strong growth condition used in Ma et al. (2018); Meng et al. (2020); Qin et al. (2022); Kim et al. (2023)) or introducing additional information sharing (e.g., gradient in Mitra et al. (2021a;b)), accurate convergence can be ensured under a constant stepsize. However, these results only prove $O(1/t)$ convergence for general convex loss functions.

Inspired by the result in Lee & Wright (2019) which proves that $o(1/t)$ convergence rate can be obtained in first-order **centralized** gradient methods by employing large stepsizes, we prove that $o(1/t)$ convergence can be achieved in **general convex** federated learning, in contrast to existing state-of-the-art algorithms—which either guarantee only $O(1/t)$ convergence (Mitra et al., 2021b; Mukherjee et al., 2023; Qin et al., 2022; Khaled et al., 2020), or rely on **additional assumptions beyond convexity** to establish $o(1/t)$ rates (Jiang et al., 2024; Kovalev et al., 2022).

The main contributions of this paper are summarized as follows:

- Under the IID condition of data distribution (also called strong growth condition in Schmidt & Roux (2013)), we prove that the conventional FedAvg algorithm (called Algorithm 1 in this paper after incorporating local stepsizes) can converge under a stepsize that is much larger than existing counterparts (our theoretically obtained stepsize is at least **two and four times larger** than the ones in Qin et al. (2022) and Khaled et al. (2020), respectively). More importantly, we prove that our stepsize can lead to an $o(1/t)$ convergence to an accurate optimal solution, faster than the commonly believed $O(1/t)$ convergence. To our knowledge, no $o(1/t)$ convergence results have been reported in the literature for **general** convex federated learning, even after incorporating momentum (see, e.g., Xu et al. (2021b); Liu et al. (2020); Cheng et al. (2023); Yang et al. (2022)).

- Under the same condition, we prove that FedAvg can converge accurately when individual clients select their (constant) stepsizes in an uncoordinated way. This allows individual clients to exploit their local geometry of loss functions and is proven in our numerical experiments to provide a faster convergence compared with the case where a same universal stepsize is used by all clients. To our knowledge, this is the first time that such results are reported for general convex loss functions.

- Under non-IID data, we show that our Algorithm 2 can ensure accurate convergence under constant stepsizes. Compared with existing counterparts, we allow much larger stepsizes (our theoretically obtained stepsize is at least $162/5$ and $4$ times larger than the ones in Karimireddy et al. (2020) and Mitra et al. (2021b), respectively). More importantly, we prove that under our stpesizes, the algorithm can ensure $o(1/t)$ convergence under a **general convex** loss function, which has not been reported before for first-order federated learning algorithms, even after incorporating momentum (Xu et al., 2021b; Liu et al., 2020; Cheng et al., 2023; Yang et al., 2022). This stands in stark contrast to existing results on convex federated learning, where $o(1/K)$ convergence has only been established under **subclasses of convex functions**—such as gradient difference being uniformly bounded (Jiang et al., 2024), or Hessian difference being uniformly bounded (Kovalev et al., 2022).

- Algorithm 2 introduces a general framework for federated learning with drift correction, unifying and extending a broad class of methods that ensure convergence under non-IID data, including FedLin (Mitra et al., 2021b), FedTrack (Mitra et al., 2021a), Scaffnew (Mishchenko et al., 2022), and SCAFFOLD (Karimireddy et al., 2020). In addition, we develop a novel analytical framework that establishes a key monotonic descent property, enabling us to prove an improved $o(1/t)$ convergence rate under **general convex** objectives—an achievement that, to the best of our knowledge, has previously only been attained for **specific subclasses** of convex functions in federated learning (see, e.g., Jiang et al. (2024); Kovalev et al. (2022)). It is worth noting that extending the monotonic descent property from centralized optimization in (Lee & Wright, 2019) to federated learning is highly nontrivial, due to the presence of multiple heterogeneous local loss functions arising from non-IID data distributions. To the best of our knowledge, this is the first work to rigorously establish such monotonicity in the context of convex federated learning.

## 2 PRELIMINARIES

**Notations** $\mathbb{R}^n$ and $\mathbb{R}^{n \times n}$ denote the set of real $n$-dimensional vectors and the set of $n \times n$-dimensional matrices, respectively. For $x \in \mathbb{R}^n$ and $A \in \mathbb{R}^{n \times n}$, $[x]_j$ and $[A]_{ij}$ denote the $j^{th}$ element of the vector $x$ and the $(i, j)^{th}$ element of the matrix $A$, respectively. For $x, y \in \mathbb{R}^n$, we define $\langle x, y \rangle = \sum_{i=1}^{n} [x]_i [y]_i$ and $\|x\| = \sqrt{\sum_{j=1}^{n} [x]_j^2}$. For a matrix $A \in \mathbb{R}^{n \times n}$, we define $\|A\|_2 = \sup_{\|x\|=1, x \in \mathbb{R}^n} \|Ax\|$. $\mathbf{0}_n \in \mathbb{R}^n$ and $\mathbf{1}_n \in \mathbb{R}^n$ are $n$-dimensional vectors with all elements being 0 and 1, respectively. We use $O(c(t))$ and $o(c(t))$ to represent sequences $d(t)$ satisfying $\limsup_{t \to +\infty} |\frac{d(t)}{c(t)}| < \infty$ and $\lim_{t \to \infty} \frac{d(t)}{c(t)} = 0$, respectively.

### 2.1 PROBLEM SETTINGS

We consider the following federated learning problem with clients set $\mathcal{S} = \{1, 2, \cdots, N\}$ as follows:

$$\min_{x \in \mathbb{R}^n} f(x) = \frac{1}{N} \sum_{i=1}^{N} f_i(x), \tag{1}$$

where $f_i : \mathbb{R}^n \to \mathbb{R}$ is the local loss function of client $i$. The local loss function $f_i(x)$ is dependent on the local training data of client $i$. We use the following standard assumptions about the loss functions (see Mitra et al. (2021a;b); Qin et al. (2022); Mukherjee et al. (2023)).

**Assumption 1.** *For any $i \in \mathcal{S}$, $f_i(x)$ is $L_i$-smooth over $\mathbb{R}^n$, i.e., there exists a constant $L_i$ such that $\|\nabla f_i(x) - \nabla f_i(y)\| \le L_i \|x - y\|$ holds for any $i \in \mathcal{S}$ and $x, y \in \mathbb{R}^n$. This implies $\|\nabla f(x) - \nabla f(y)\| \le L\|x - y\|$ where $L = \frac{1}{N} \sum_{i=1}^{N} L_i$, i.e., $f(x)$ is also $L$-smooth over $\mathbb{R}^n$.*

**Assumption 2.** *For any $i \in \mathcal{S}$, $f_i(x)$ is convex over $\mathbb{R}^n$. Moreover, the optimal solution set $\mathcal{X}^* = \{x^* \in \mathbb{R}^n | x^* = \arg\min_{x \in \mathbb{R}^n} f(x)\}$ is not empty, i.e., there exists at least one $x^* \in \mathbb{R}^n$ such that $\nabla f(x^*) = \mathbf{0}_n$ holds.*

In existing results for federated learning (see, e.g., Mitra et al. (2021b); Qin et al. (2022); Khaled et al. (2020); Mukherjee et al. (2023)), the theoretically obtained convergence rates are all on the order of $O(1/t)$ for general convex loss functions, where $t$ is the number of communications between clients and the central server. In this paper, we will show that we can prove a faster $o(1/t)$ convergence rate by using a larger stepsize. To this end, we first introduce the following lemma (see Debnath & Mikusinski (2005) or Lee & Wright (2019)).

**Lemma 1.** *Let $\{\Delta(t)\}$ be a nonnegative sequence satisfying the following conditions:*

*(1) $\{\Delta(t)\}$ is monotonically decreasing;*

*(2) $\{\Delta(t)\}$ is summable, that is, $\sum_{k=0}^{\infty} \Delta(k) < \infty$.*

*Then, we have $\Delta(t) = o(1/t)$, i.e., $\lim_{t \to \infty} t\Delta(t) = 0$.*

## 3 CONVERGENCE UNDER IID DATA

In this section, we consider the case where the data on all clients are IID. In the literature, this is usually formulated as the following assumption (see, e.g., Schmidt & Roux (2013); Qin et al. (2022); Kim et al. (2023)):

**Assumption 3.** *There exists a constant $\eta > 0$ such that $\|\nabla f_i(x)\| \le \eta \|\nabla f(x)\|$ holds for any client $i \in \mathcal{S}$ and $x \in \mathbb{R}^n$.*

This assumption is also sometimes called Strong Growth Condition (Schmidt & Roux, 2013) and has been widely used in machine learning (Ma et al., 2018; Vaswani et al., 2019a;b; Gower et al., 2021; Meng et al., 2020). In fact, Qin et al. (2022) recently experimentally verified that this condition is satisfied for over-parameterized models. Next, we will prove that the classic federated learning algorithm FedAvg can converge at an $o(1/t)$ rate under judiciously designed stepsizes under Assumption 3. In Section 4, we will consider the more general non-IID case.

### 3.1 ALGORITHM DESCRIPTION

For the sake of completeness, we restate FedAvg in McMahan et al. (2017) as Algorithm 1 (with an extension that we allow clients to use heterogeneous local stepsizes). Specifically, in this algorithm, instead of using a universal stepsize $\alpha$, each client selects its own stepsize $\alpha_i$ without coordination with other clients. As proven in the next subsection and the numerical experimental evaluation, this enables our algorithm to obtain faster convergence than existing counterparts.

---

**Algorithm 1** (FedAvg with local stepsizes)

   **Input**: Initial value $\bar{x}(1)$, local training period $\tau$, the stepsize $\alpha_i$ for client $i$
   **for** $t = 1$ **to** $T$ **do**
     **for** $i = 1$ **to** $N$ **do**
       Each client $i$ sets $x_{i,0}(t) = \bar{x}(t)$.
       **for** $k = 0$ **to** $\tau - 1$ **do**
         Each client $i$ does local training

$$x_{i,k+1}(t) = x_{i,k}(t) - \alpha_i \nabla f_i(x_{i,k}(t)). \tag{2}$$

       **end for**
     **end for**
     Each client $i$ transmits $x_{i,\tau}(t)$ to the central server and receives $\bar{x}(t+1) = \frac{1}{N} \sum_{i=1}^{N} x_{i,\tau}(t)$ from the central server.
   **end for**

---

### 3.2 CONVERGENCE ANALYSIS

**Theorem 1.** *Under Assumptions 1, 2, and 3, if the stepsize of client $i$ satisfies $\alpha_i = \alpha > 0$ for all $i \in \mathcal{S}$ and*

$$\alpha < \min_{1 \leq i \leq N} \left\{ \frac{1}{L_i \tau}, \frac{8\tau}{L(2\tau + \eta(\tau - 1))^2 + 4\eta L \tau(\tau - 1)} \right\}, \tag{3}$$

*where $L = \frac{1}{N} \sum_{i=1}^{N} L_i$, then $f(\bar{x}(t))$ converges to $f(x^*)$ with the convergence rate $o(1/t)$, i.e.,*

$$\lim_{t \to \infty} t\{f(\bar{x}(t)) - f(x^*)\} = 0.$$

*Proof.* See Appendix C.     □

In fact, we can allow the stepsize $\alpha_i$ of the client $i \in \mathcal{S}$ in Theorem 1 to be larger to achieve faster convergence of Algorithm 1, which is detailed in Theorem 2.

**Theorem 2.** *Under Assumptions 1, 2, and 3, if the stepsize $\alpha_i$ for client $i \in \mathcal{S}$ in Algorithm 1 satisfies*

$$0 < \alpha_i < \frac{1}{L_i}, \tag{4}$$

*we have $\lim_{t \to \infty} f(\bar{x}(t)) = f(x^*)$ and $f(\frac{1}{T} \sum_{t=1}^{T} \bar{x}(t)) - f(x^*) \leq \frac{\|\bar{x}(1) - x^*\|^2}{\min_{1 \leq i \leq N}\{2\alpha_i - 2L_i \alpha_i^2\}T}$.*

*Proof.* See Appendix D.     □

The proposed stepsize in Theorem 2 is larger than designed stepsizes for FedAvg in existing theoretical results. For example, Qin et al. (2022) and Khaled et al. (2020) obtained stepsizes that should satisfy $0 < \alpha \leq \frac{1}{2L}$ and $0 < \alpha \leq \frac{1}{4L}$, respectively. A simple comparison with (4) shows that our stepsize can be **two and four times as large** besides the additional flexibility of allowing different clients to select their local stepsizes to exploit local geometry to speed up convergence. In fact, our numerical experiments in Figure 1 confirm that our stepsize strategy indeed leads to much faster convergence than the ones in Qin et al. (2022); Khaled et al. (2020); Mukherjee et al. (2023) (see Table 1 for a detailed comparison of stepsizes).

Theorem 2 can also be obtained under a weaker interpolation assumption: $\|\nabla f_i(x^*)\| = 0$ for all $i \in \mathcal{S}$, $x \in \mathbb{R}^n$, and $x^* \in \mathcal{X}^*$, which is also widely investigated in machine learning (see Ma et al. (2018); Vaswani et al. (2019a;b); Gower et al. (2021); Meng et al. (2020)). Compared with Theorem 2, Theorem 2 does not require client $i$ to know information about the global loss function to determine its stepsize. In addition, it allows stepsize that is $\max_{1 \le j \le N}\{\frac{L_j\tau}{L_i}, \frac{L(2+\eta(\tau-1))^2+4\eta L\tau(\tau-1)}{8L_i\tau}\}$ times larger than that in (3). In fact, our numerical experimental results in Appendix A.2.2 show that allowing clients to use local stepsizes achieves a faster convergence than the case with a global stepsize. This is intuitive in that utilizing local Lipschitz constants allows the gradient descent steps to exploit the local geometry of loss functions, and hence, enables faster convergence. It is worth noting that although such a phenomenon has been reported in Mukherjee et al. (2023) for one specific example of quadratic functions, we are the first to theoretically establish that local stepsizes can be exploited to achieve faster convergence for a general class of loss functions in federated learning.

**Remark 1.** *It is worth noting that the $o(1/k)$ convergence rate established in Theorem 1 does not contradict the result in Glasgow et al. (2022), which proves that FedAvg cannot achieve a rate faster than $O(1/k)$ for general convex objectives. The key distinction lies in the fact that Theorem 1 relies on the additional Strong Growth Condition (see Assumption 3). In the next section, we introduce a new algorithm that achieves $o(1/k)$ convergence for general convex objectives under non-IID data, without requiring any additional restrictive conditions.*

We can extend our analysis to the setting of stochastic gradients, where client $i$ can only access an unbiased estimate of the gradient $\nabla f_i(x)$ with variance bounded by $\sigma^2$. More specifically, Corollary 3 (see proof in Appendix E) establishes the convergence of Algorithm 1 in this case:

**Corollary 3** (Stochastic Gradients)**.** *Under Assumptions 1, 2, and 3, if the stepsize $\alpha_i$ in Algorithm 1 satisfies $0 < \alpha_i < \frac{1}{L_i}$ for any $i \in \mathcal{S}$, we have*

$$\mathbb{E}\Big[f\Big(\frac{1}{T}\sum_{t=1}^{T}\bar{x}(t)\Big)\Big] - f(x^*) \le \frac{\|\bar{x}(1) - x^*\|^2}{\min_{1 \le i \le N}\{2\alpha_i - 2L_i\alpha_i^2\}T} + \frac{2\tau\sum_{i=1}^{N}\alpha_i^2}{\min_{1 \le i \le N}\{2\alpha_i - 2L_i\alpha_i^2\}N}\sigma^2.$$

## 4 CONVERGENCE UNDER NON-IID DATA

### 4.1 ALGORITHM DESCRIPTION

Under non-IID data, it has been known that except the trivial case where the number of local iterations is one ($\tau = 1$), Algorithm 1 will be subject to errors (Mukherjee et al., 2023; Orvieto et al., 2022; Wang et al., 2020; Karimireddy et al., 2020). Inspired by gradient-tracking-based distributed optimization algorithms (Pu & Nedić, 2021; Nedić et al., 2017), we propose Algorithm 2 to address this issue and ensure accurate convergence under non-IID data.

Unlike Algorithm 1 which exchanges only the model parameters $x_{i,k+1}(t)$ between clients and the server, Algorithm 2 requires exchanging an additional variable for the gradient. More specifically, in Algorithm 2, each client uses the global gradient information $\nabla f(\bar{x}(t))$ to initialize its local variable $y_{i,k}(t)$ after each communication round (see (5)). This variable $y_{i,k+1}(t)$, which serves as an estimate of the global gradient, is then used to update the model parameter $x_{i,k+1}(t)$ (see (7)). This is key to eliminating the drift caused by non-IID data.

Our Algorithm 2 provides a general framework for federated learning with drift correction, encompassing a wide range of existing algorithms as special cases. Specifically, by substituting equation (5) into equation (7) and applying mathematical induction, the auxiliary variable $y_{i,k}(t)$ can be expressed as $y_{i,k}(t) = \nabla f(x_{i,k}(t)) - \nabla f_i(\bar{x}(t)) + \nabla f(\bar{x}(t))$ for $k = 0, 1, \ldots, \tau$. Substituting this expression into the update rule (6) recovers the specific update mechanisms used in FedLin (Mitra et al., 2021b) and FedTrack (Mitra et al., 2021a). In addition, as $x_{i,k}(t)$ converges to $x^*$, it follows from equation (8) that $y_{i,k}(x^*) = \nabla f_i(x^*)$, a key idea leveraged in the "drift-correction" federated learning algorithms Scaffnew (Mishchenko et al., 2022) and SCADDOLD (Karimireddy et al., 2020). This demonstrates that Algorithm 2 not only generalizes but also unifies prior drift-corrected federated learning methods within a broader and more flexible structure.

Next, we prove that the new framework allows us to obtain $o(1/t)$ convergence in general convex federated learning, which is only established in the literature for special classes of convex functions

---

**Algorithm 2**

---

**Input:** Initial values $\bar{x}(1)$, $\nabla f(\bar{x}(1))$, local training period $\tau$, and stepsize $\alpha$;

**for** $t = 1$ **to** $T$ **do**

    **for** $i = 1$ **to** $N$ **do**

        Each client $i$ sets

$$x_{i,0}(t) = \bar{x}(t) \quad \text{and} \quad y_{i,0}(t) = \nabla f(\bar{x}(t)). \tag{5}$$

        **for** $k = 0$ **to** $\tau - 1$ **do**

            Client $i$ does local updating

$$x_{i,k+1}(t) = x_{i,k}(t) - \alpha y_{i,k}(t), \tag{6}$$

$$y_{i,k+1}(t) = y_{i,k}(t) + \nabla f_i(x_{i,k+1}(t))) - \nabla f_i(x_{i,k}(t))). \tag{7}$$

        **end for**

    **end for**

    The central server calculates and transmits $\bar{x}(t + 1) = \frac{1}{N} \sum_{i=1}^{N} x_{i,\tau}(t)$ to each client. Each client $i$ then transmits $\nabla f_i(\bar{x}(t + 1))$ to the central server and receives $\nabla f(\bar{x}(t + 1)) = \frac{1}{N} \sum_{i=1}^{N} \nabla f_i(\bar{x}(t + 1))$ from the central server.

**end for**

---

with restrictions on data heterogeneity (see, e.g., under the bounded gradient difference condition in Jiang et al. (2024) and under the bounded Hessian difference condition in Kovalev et al. (2022)). For the general convex case without any restrictions, existing federated learning algorithms—even those incorporating momentum—only achieve an $O(1/t)$ convergence rate. In addition, the new framework allows using significantly larger step sizes compared to existing drift-corrected federated learning algorithms, as detailed in Section 4.2.

## 4.2 CONVERGENCE ANALYSIS

**Theorem 4.** *For Algorithm 2, under Assumptions 1 and 2, if the stepsize $\alpha$ of client $i \in \mathcal{S}$ satisfies*

$$0 < \alpha < \min_{1 \leq j \leq N} \left\{ \frac{1}{L_j}, \frac{2}{5L\tau - L} \right\}, \tag{8}$$

*where $L = \frac{1}{N} \sum_{i=1}^{N} L_i$, then $f(\bar{x}(t))$ converges to $f(x^*)$ with the convergence rate $o(1/t)$, i.e.,*

$$\lim_{t \to \infty} t\{f(\bar{x}(t)) - f(x^*)\} = 0.$$

*Proof.* See Appendix F. $\qquad\square$

In Theorem 4, we establish an $o(1/t)$ convergence rate for federated learning with general convex functions under non-IID data. A key step in this analysis, as shown in Lemma 1, is proving that the sequence $\{f(\bar{x}(t))\}$ is monotonically decreasing, i.e.,

$$f(\bar{x}(t + 1)) \leq f(\bar{x}(t)).$$

We emphasize that proving this monotonicity under general smooth and convex conditions is highly nontrivial. Our proof of this property, presented in Lemma 5, constitutes a significant technical contribution of this work.

Notably, other federated learning algorithms in Mitra et al. (2021a;b), which also follow a gradient-tracking-based framework, only establish an $O(1/t)$ convergence rate under general convex functions in their analyses. In contrast, our work develops a more refined analysis technique—specifically, the nontrivial proof of the monotonically decreasing property, i.e., $f(\bar{x}(t+1)) \leq f(\bar{x}(t))$ (see Lemma 5)—which enables us to establish an $o(1/t)$ convergence rate in Theorem 4. Importantly, this analysis framework is not limited to our algorithm and can also be applied to other gradient-tracking-based methods to improve their theoretical guarantees from $O(1/t)$ to $o(1/t)$ under general convex settings. This general methodology, therefore, represents a significant contribution of our work.

In the case of stochastic gradients, client $i$ can only access an unbiased estimate of the gradient $\nabla f_i(x)$ with variance bounded by $\sigma^2$. Next, we establish Corollary 5 (see proof in Appendix H) for the convergence of Algorithm 2 in this stochastic setting:

**Corollary 5** (Stochastic Gradients)**.** *Under Assumptions 1 and 2, if the stepsize $\alpha$ of Algorithm 2 satisfies $0 < \alpha \leq \min_{1 \leq j \leq N}\{\frac{1}{L_j}, \frac{1}{12\tau L}\}$, we have*

$$\mathbb{E}\Big[f\Big(\frac{1}{T}\sum_{t=1}^{T}\bar{x}(t)\Big)\Big] - f(x^*) \leq \frac{\|x(1) - x^*\|^2}{\alpha\tau T} + 34\tau\alpha\sigma^2.$$

### 4.3 COMPARISON WITH EXISTING RESULTS

From Theorem 4, Algorithm 2 allows a much larger stepsize and a better convergence rate compared with existing works. Specifically, the stepsize in Karimireddy et al. (2020) is required to satisfy $0 < \alpha \leq \min_{1 \leq i \leq N}\{\frac{1}{81 L_i \tau}\}$. In contrast, the stepsize upper bound in Theorem 4 is given by $\min_{1 \leq i \leq N}\{\frac{1}{L_i}, \frac{2}{5L\tau - L}\}$. It can be verified that our permissible stepsize is at least $\frac{162}{5}$ times larger than that in Karimireddy et al. (2020). Similarly, Mitra et al. (2021b) requires the stepsize to satisfy $0 < \alpha \leq \min_{1 \leq i \leq N}\{\frac{1}{10 L_i \tau}\}$. In contrast, our Theorem 4 permits a stepsize that is at least $\max_{1 \leq i \leq N}\{\frac{20\tau L_i}{5L\tau - L}\} \geq 4$ times larger than that in Mitra et al. (2021b). Table 1 provides a detailed comparison between our proposed stepsize and convergence rate with existing works.

Table 1: Comparison of the proposed stepsizes and obtained convergence rates for Algorithm 1 and Algorithm 2 with existing results. In this table, we represent the total communication round as $t$, the local training period as $\tau$, and assume that the local loss function $f_i(x)$ satisfies $L$-smooth property and each client uses precise gradient.

| ASSUMPTION | ALGORITHM | STEPSIZE | CONVERGENCE RATE |
|---|---|---|---|
| IID | ALGORITHM 1 | $1/L$ | $O(1/t)$ |
| | QIN ET AL. (2022) | $1/(2L)$ | $O(1/t)$ |
| | KHALED ET AL. (2020) | $1/(4L)$ | $O(1/t)$ |
| NON-IID | ALGORITHM 2 | $2/(5L\tau - L)$ | $o(1/t)$ |
| | MITRA ET AL. (2021B); KHALED ET AL. (2020) | $1/(10\tau L)$ | $O(1/K)$ |
| | MITRA ET AL. (2021A) | $1/(18\tau L)$ | $O(1/K)$ |
| | KARIMIREDDY ET AL. (2020) | $1/(81\tau L)$ | $O(1/K)$ |
| | REISIZADEH ET AL. (2020); ZHU ET AL. (2021) XIANG ET AL. (2024); HUANG ET AL. (2023) WANG ET AL. (2020); YU ET AL. (2019) YANG ET AL. (2021); LI & LI (2023) HADDADPOUR & MAHDAVI (2019) | $O(1/\sqrt{K})$ | $O(1/\sqrt{K})$ |
| | KIM ET AL. (2023) | ADAPTIVE | $O(1/\sqrt{K})$ |

## 5 EXPERIMENTS

### 5.1 EVALUATION USING GENERATED DATA UNDER IID DISTRIBUTION

We use the following regression problem to compare the performance of Algorithm 1 and Algorithm 2 under the proposed stepsizes with existing counterparts[1]:

$$\min_{x \in \mathbb{R}^n} f(x) = \min_{x \in \mathbb{R}^n} \frac{1}{N}\sum_{i=1}^{N}\frac{1}{2}\|A_i x - b_i\|^2, \tag{9}$$

where $A_i \in \mathbb{R}^{500 \times 100}$, $b_i \in \mathbb{R}^{500}$ and $x \in \mathbb{R}^{100}$ for each client $i \in \mathcal{S} = \{1, 2, \cdots, 20\}$. $[A_i]_{jk}$ are generated from $[0, 1]$ randomly for $1 \leq j \leq 500$, $1 \leq k \leq 100$, and $i \in \mathcal{S}$, and we also set $[A_1]_{j,1} =$

---

[1]Code available at https://anonymous.4open.science/r/o1_t-F814/README.md

$[A_1]_{j,2}$ for $1 \leq j \leq 500$ to obtain a convex but not strongly convex loss function $f_1(x)$. We set $b_i = A_i x_0$ for all $i \in \mathcal{S}$ with $x_0 = 10 \times \mathbf{1}_n$ rather than generating $b_i$ randomly. In this setting, $f_i(x) = \frac{1}{2}\|A_i(x - x_0)\|^2$ and, hence, there exists a constant $\eta = \max_{1 \leq j \leq N}\left\{\frac{\|A_j^T A_j\|_2}{\|(\frac{1}{N}\sum_{i=1}^{N} A_i)^T(\frac{1}{N}\sum_{i=1}^{N} A_i)\|_2}\right\}$ such that $\|\nabla f_i(x)\| \leq \eta\|\nabla f(x)\|$ holds for all $i \in \mathcal{S}$.

We compare Algorithm 1 and Algorithm 2 under the proposed stepsize strategy with existing counterparts including Qin et al. (2022); Mukherjee et al. (2023); Mitra et al. (2021b); Khaled et al. (2020). In the evaluation, we use the error $e(t) = f(\bar{x}(t)) - f(x^*)$ to measure the learning accuracy. Moreover, we implement all algorithms using accurate gradients to ensure a fair comparison of them. The corresponding convergence performances with different local training periods $\tau = 2, 3, 4, 5, 6$ are presented in Figure 1.

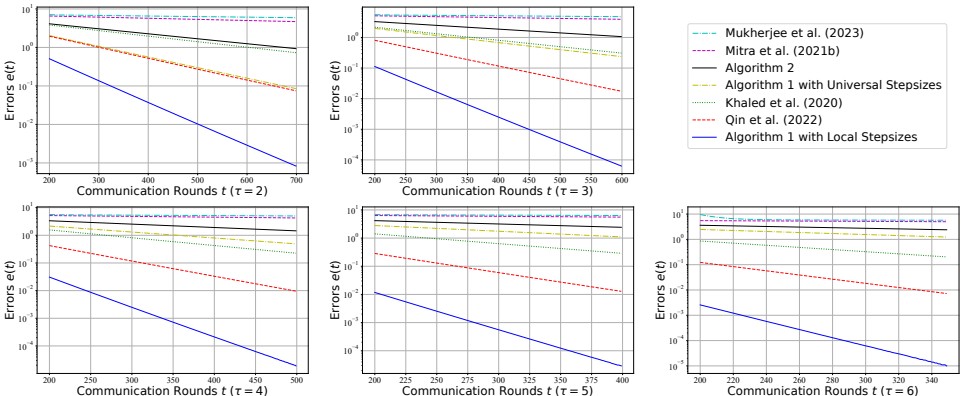

Figure 1: Comparisons of the performance of Algorithm 1 and Algorithm 2 under the proposed stepsize with Qin et al. (2022); Mukherjee et al. (2023); Mitra et al. (2021b); Khaled et al. (2020) under different local training periods $\tau$.

In Figure 1, the legends 'Algorithm 1 with Universal Stepsizes' and 'Algorithm 1 with Local Stepsizes' denote Algorithm 1 with stepsizes (3) and (4), respectively. Specifically, in the universal stepsize case, we set the universal stepsize $\alpha$ for all clients as $\alpha = \min_{1 \leq i \leq N}\left\{\frac{1}{L_i\tau}, \frac{8\tau}{L(2\tau+\eta(\tau-1))^2+4\eta L\tau(\tau-1)}\right\} - 10^{-15}$ according to (3), where $L = \frac{1}{N}\sum_{i=1}^{N} L_i$ is the global Lipschitz constant. In the local stepsize case, we set the stepsize of client $i$ as $\alpha_i = \frac{1}{L_i} - 10^{-15}$ based on individual Lipschitz constants $L_i$. From Figure 1, we know that the convergence of Algorithm 1 with the stepsize prescribed in (4) is much faster than other cases, including the case with the universal stepsize (3). Additional experiments with non-IID data are presented in Appendix A.2.

## 5.2 EVALUATION USING CIFAR-10 AND CIFAR-100 UNDER NON-IID DISTRIBUTION

We also evaluate our algorithms by training a CNN on 10 clients using the benchmark datasets CIFAR-10 and CIFAR-100, respectively[2]. The CNN architecture consists of three convolutional layers with 32, 64, and 128 filters, respectively, each followed by a max-pooling layer. After the final convolutional and pooling layers, the network includes a fully connected layer with 256 units and ReLU activation, a dropout layer with a rate of 0.25 for regularization, and a final dense output layer with 10 units that produces the class logits. In our experiments, we compare the proposed algorithm against existing federated learning methods specifically designed to address client drift, including SCAFFOLD (Karimireddy et al., 2020), FedLin (Mitra et al., 2021b), and Scaffnew (Mishchenko et al., 2022). Following Hsu et al. (2019) and Kim et al. (2023), we generate heterogeneous data distributions across the 10 agents using a Dirichlet distribution, with the heterogeneity parameter $\alpha$ set to 0.1, 1, and 10, respectively. A higher value of $\alpha$ yields a nearly uniform distribution of data across classes for each client, resulting in approximately IID local datasets. In contrast, a lower $\alpha$ leads to highly skewed distributions, where clients tend to specialize in only a few classes.

---

[2]Code available at https://anonymous.4open.science/r/o1_t-F814/README.md

Figures 2 and 3 report results for $\alpha = 1$, which corresponds to a moderately heterogeneous setting (additional results for other values of $\alpha$ are provided in Appendix A.1). In both Figure 2 (CIFAR-10) and Figure 3 (CIFAR-100), the step sizes for Algorithm 2, SCAFFOLD, FedLin, and Scaffnew are selected according to the guidelines from Theorem 4, Karimireddy et al. (2020), Mitra et al. (2021b), and Mishchenko et al. (2022), respectively, using an estimated smoothness parameter of $L = 2$. For Algorithm 2, SCAFFOLD, and FedLin, the local training period is set to $\tau = 10$. For Scaffnew, the communication probability is set to $\frac{1}{11}$ to ensure that the total number of communicated messages remains consistent across methods. As shown in the figures, our algorithm achieves faster convergence and higher accuracy on both the CIFAR-10 dataset and the CIFAR-100 dataset.

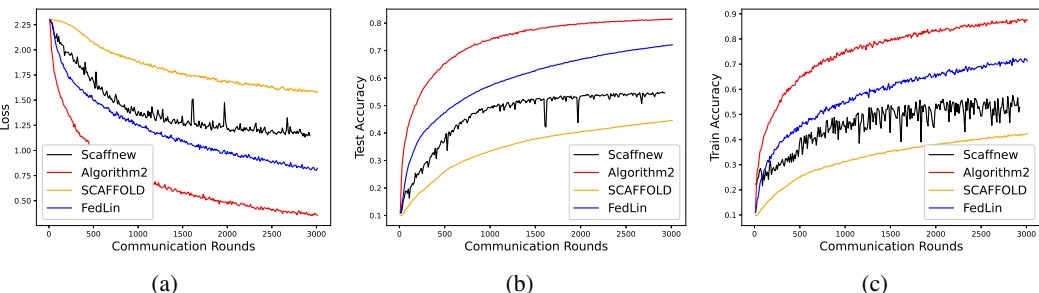

       (a)                 (b)               (c)

Figure 2: Comparison of Algorithm 2 with state-of-the-art federated learning algorithms—SCAFFOLD, FedLin, and Scaffnew—on the CIFAR-10 dataset. Each curve represents the average of five independent runs.

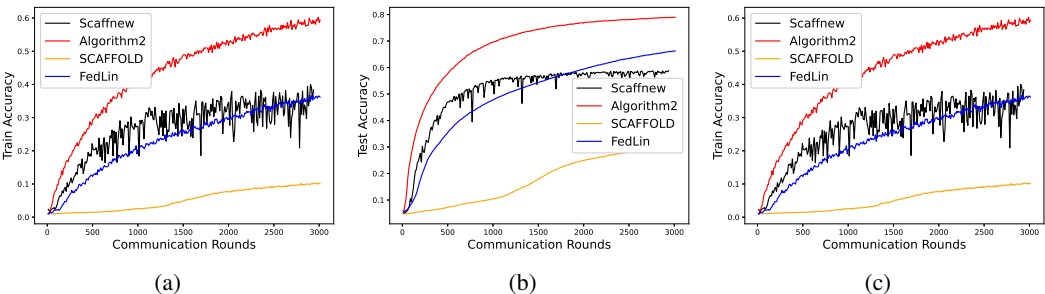

       (a)                 (b)               (c)

Figure 3: Comparison of Algorithm 2 with state-of-the-art federated learning algorithms—SCAFFOLD, FedLin, and Scaffnew—on the CIFAR-100 dataset. Each curve represents the average of five independent runs. The test accuracy in Figure 2(b) is top-5 accuracy.

## 6 CONCLUSION

Enhancing convergence accuracy and speed is key for federated learning. We prove that much larger stepsizes can be used in FedAvg, and hence, much faster convergence can be achieved. In fact, we theoretically show that the proposed stepsize strategy can guarantee $o(1/t)$ convergence to an exact optimal solution for general convex loss functions, under both IID data distribution and non-IID data distribution among local clients. This is significant since existing federated learning results can only theoretically establish $O(1/t)$ convergence under general convex loss functions when no additional restrictions are made, even after incorporating momentum. Moreover, in the IID data distribution setting, we theoretically establish convergence when clients set stepsizes individually using local Lipschitz parameters, and show that such a local stepsize strategy enables exploiting local geometry to expedite convergence. To our knowledge, this is the first time that local stepsizes designed using local Lipschitz parameters is systemtically shown to outperform a universal stepsize designed using the global Lipschitz parameter. Moreover, we propose a general gradient-tracking-based framework that unifies and extends many existing drift-corrected federated learning algorithms. By establishing a key monotonic descent property, our framework broadens the theoretical understanding of gradient tracking and enables an improved $o(1/t)$ convergence rate under non-IID data distributions. This represents a significant advancement, as existing results establish $o(1/t)$ convergence for convex federated learning only under additional restrictions on heterogeneity.

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

## A ADDITIONAL NUMERICAL EXPERIMENTS

### A.1 ADDITIONAL CNN TRAINING RESULTS WITH DIFFERENT NON-IID LEVELS

Figures 4 and 5 provide additional results for the CNN training experiment in Section 5.2 with a heterogeneity parameter $\alpha = 0.1$.

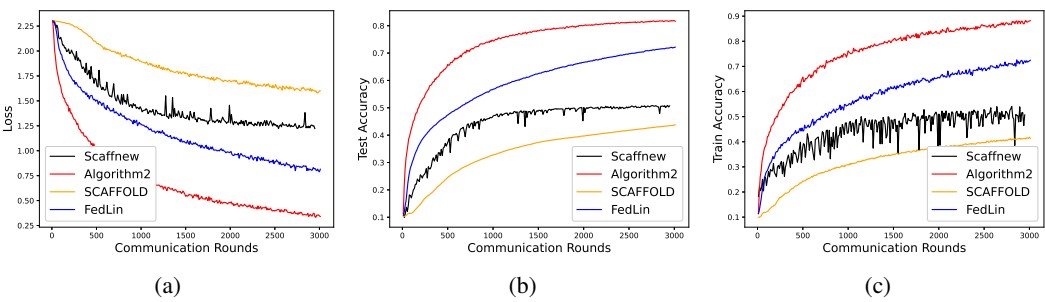

|   (a)   |   (b)   |   (c)   |

Figure 4: Comparison of Algorithm 2 with state-of-the-art federated learning algorithms—SCAFFOLD, FedLin, and Scaffnew—on the CIFAR-10 dataset. Each curve represents the average of five independent runs. To induce greater heterogeneity in data distribution, the Dirichlet distribution parameter was set to $\alpha = 0.1$.

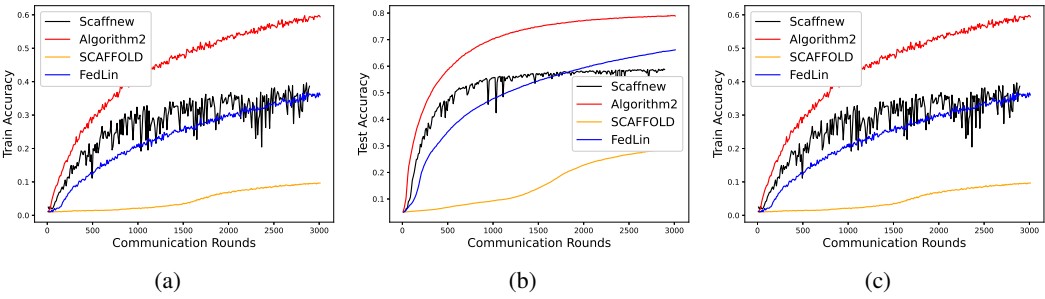

|   (a)   |   (b)   |   (c)   |

Figure 5: Comparison of Algorithm 2 with state-of-the-art federated learning algorithms—SCAFFOLD, FedLin, and Scaffnew—on the CIFAR-100 dataset. Each curve represents the average of five independent runs. The test accuracy in Figure 2(b) is top-5 accuracy. To induce greater heterogeneity in data distribution, the Dirichlet distribution parameter was set to $\alpha = 0.1$.

Figures 6 and 7 provide additional results for the CNN training experiment in Section 5.2 with a heterogeneity parameter $\alpha = 10$.

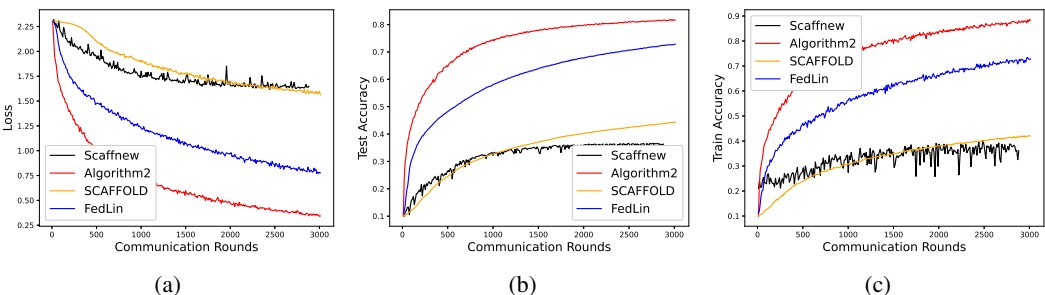

Figure 6: Comparison of Algorithm 2 with state-of-the-art federated learning algorithms—SCAFFOLD, FedLin, and Scaffnew—on the CIFAR-10 dataset. Each curve represents the average of five independent runs. To induce smaller heterogeneity in data distribution, the Dirichlet distribution parameter was set to $\alpha = 10$.

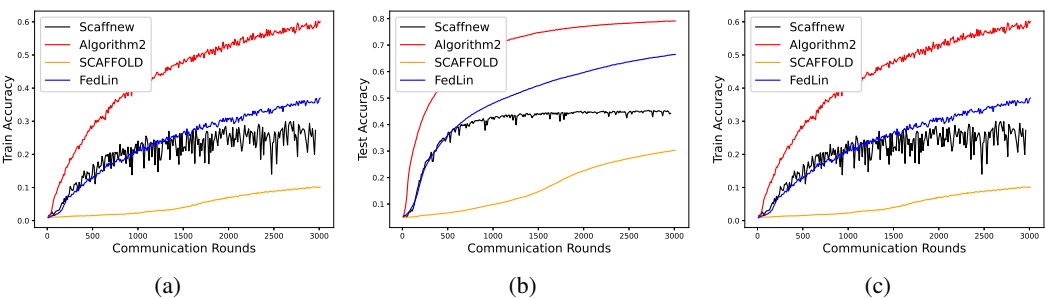

Figure 7: Comparison of Algorithm 2 with state-of-the-art federated learning algorithms—SCAFFOLD, FedLin, and Scaffnew—on the CIFAR-100 dataset. Each curve represents the average of five independent runs. The test accuracy in Figure 2(b) is top-5 accuracy. To induce smaller heterogeneity in data distribution, the Dirichlet distribution parameter was set to $\alpha = 10$.

### A.2 LEAST SQUARES REGRESSION

#### A.2.1 COMPARISON OF ALGORITHM 2 WITH EXISTING WORKS

We consider $[A_i]_{jk}$ and $[b_i]_j$ generated from $[0, 1]$ randomly for $1 \le j \le 500$, $1 \le k \le 100$, and $i \in \mathcal{S}$. After the initial random generation of data, we purposely set $[A_1]_{j,1} = [A_1]_{j,2}$ for $1 \le j \le 500$ to make $A_1$ not full rank. By doing so, we can obtain a local loss function $f_1(x) = \frac{1}{2}\|A_1 x - b_1\|^2$ that is convex but not strongly convex.

We compared the performance of Algorithm 2 under the proposed stepsize (8) in Theorem 4 with those in Mitra et al. (2021a;b). The convergence performances of Algorithm 2 and algorithms in Mitra et al. (2021a;b) under different local training periods $\tau = 2, 3, 4, 5, 6, 7$ are shown as Figure 8. It is clear that the proposed stepsize strategy indeed yields much faster convergence than the compared counterparts.

#### A.2.2 LOCAL STEPSIZE STRATEGY OUTPERFORMS UNIVERSAL STEPSIZE STRATEGY FOR ALGORITHM 1 UNDER $\tau = 1$

We show that better convergence performance of Algorithm 1 can be achieved with local stepsizes $0 < \alpha_i < \frac{1}{L_i}$ than a universal stepsize $0 < \alpha \le \frac{1}{L}$, where $L = \frac{1}{N}\sum_{i=1}^{N} L_i$. For ease of comparison, we select $[A_i]_{jk} = i^\rho [B_i]_{jk}$ and $b_i = A_i x_0$ for $1 \le j \le 500$, $1 \le k \le 100$, and $i \in \mathcal{S}$, where $[B_i]_{jk}$ is generated from $[0, 1]$ randomly, $\rho$ measures the heterogeneity in loss functions, and $x_0 = 10 \times \mathbf{1}_n$. It can be seen that a larger parameter $\rho$ leads to more heterogeneity in the local loss functions. Moreover, one can verify that the loss function $f_i(x)$ of client $i \in \mathcal{S}$ satisfies $L_i$-smooth property with $L_i = i^\rho \|B_i^T B_i\|_2$. Then, under $\tau = 1$, we present in Figure 9 the convergence of Algorithm 1 under the local stepsize strategy where $\alpha_i = \frac{1}{L_i} - 10^{-15}$ of client $i \in \mathcal{S}$ and the universal stepsize

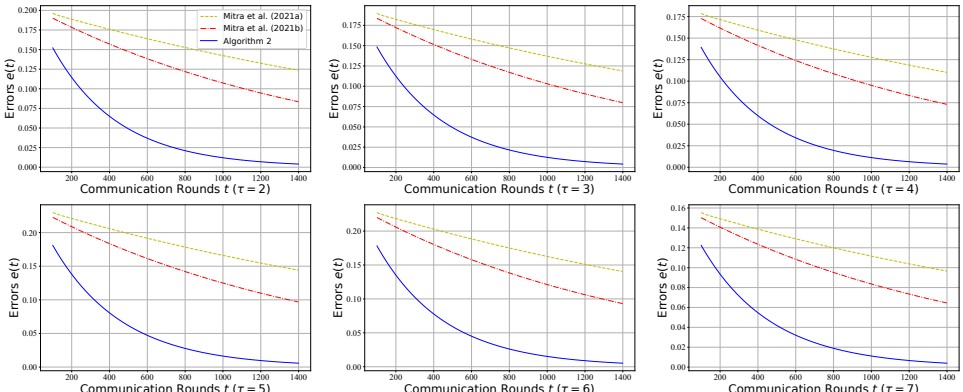

Figure 8: Comparisons of the performance of Algorithm 2 under the proposed stepsize with Mitra et al. (2021a;b) under different local training periods $\tau$

strategy where $\alpha = \frac{1}{L}$ for all clients, where $L_i$ is the individual Lipschitz constant of client $i \in \mathcal{S}$ and $L = \frac{1}{N} \sum_{i=1}^{N} L_i$ is the global Lipschitz constant.

In Figure 9, to compare the convergence between the local and the universal stepsize strategies, we plot the learning errors $f(\bar{x}_l(t)) - f(x^*)$ and $f(\bar{x}_g(t)) - f(x^*)$ under different heterogeneity parameters $\rho = 1, 1.5, 2, 2.5, 3$, where $\bar{x}_l(t)$ and $\bar{x}_g(t)$ are generated under the local and the universal stepsize strategies, respectively. From Figure 9, it is clear that the local stepsize designed based on local Lipschitz constants obtains faster convergence than the case with the universal stepsize designed based on the global Lipschitz constant. Moreover, to quantify the improvement in convergence speed, in Figure 9, we also plot the learning error ratio $r(t) = \frac{f(\bar{x}_l(t)) - f(x^*)}{f(\bar{x}_g(t)) - f(x^*)}$ under different heterogeneity parameters $\rho = 1, 1.5, 2, 2.5, 3$, respectively. A smaller $r(t)$ ($r(t) < 1$) means more advantage of the convergence speed of the local stepsize strategy over the universal stepsize strategy. Figure 9 shows that a smaller $r(t)$ is obtained under a larger heterogeneity parameter $\rho$. Thus, it can be concluded that the local stepsize strategy of Algorithm 1 can achieve faster convergence than the global stepsize strategy, especially for large heterogeneity cases.

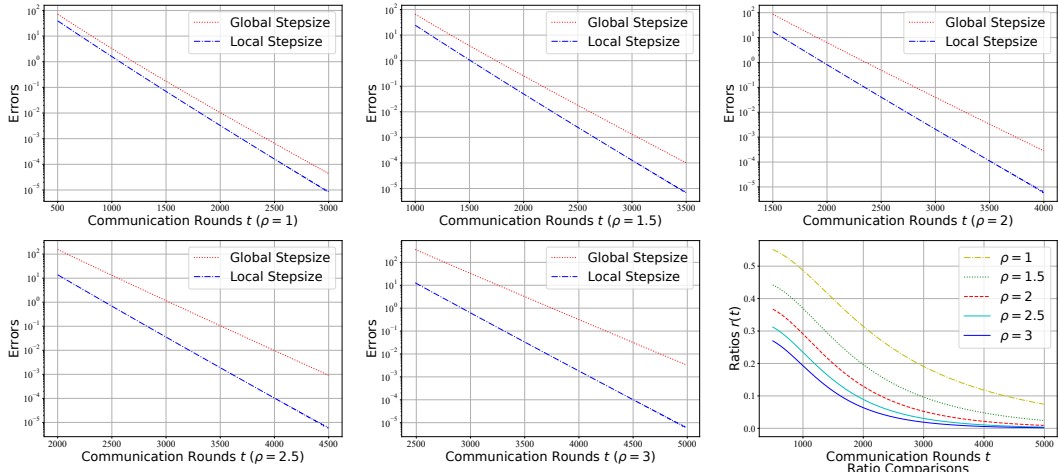

Figure 9: Comparisons of the local stepsize strategy with a universal stepsize strategy under different heterogeneity parameters $\rho$.

## B  SUPPORTING LEMMAS FOR THE PROOF OF THEOREM 1

**Lemma 2** (Zhou (2018)). *For every $L_i$-smooth and convex function $f_i(x)$ over $\mathbb{R}^n$, we have*

$$f_i(y) \geq f_i(x) + \langle \nabla f_i(x), y - x \rangle + \frac{1}{2L_i} \|\nabla f_i(x) - \nabla f_i(y)\|^2$$

*for any $x, y \in \mathbb{R}^n$ and $i \in \mathcal{S}$.*

**Lemma 3** (Mitra et al. (2021b)). *Suppose that $f_i(x)$ is $L_i$-smooth and convex. Then, for any $0 \leq \alpha \leq \frac{1}{L_i}$, we have*

$$\|y - x - \alpha(\nabla f_i(y) - \nabla f_i(x))\| \leq \|y - x\|$$

*for any $x, y \in \mathbb{R}^n$.*

**Lemma 4** (Zhou (2018)). *For the convex and $L$-smooth function $f(x)$, we have*

$$f(y) \leq f(x) + \langle \nabla f(x), y - x \rangle + \frac{L}{2} \|y - x\|^2$$

*for any $x, y \in \mathbb{R}^n$.*

## C  PROOF OF THEOREM 1

*Proof.* The sequence $\{f(\bar{x}(t)) - f(x^*)\}$ satisfies

$$f(\bar{x}(t)) - f(x^*) \geq 0$$

for any $t \geq 1$. From Lemma 1, to prove Theorem 1, we only need to prove that the nonnegative sequence $\{f(\bar{x}(t)) - f(x^*)\}$ satisfies the summable and monotonically decreasing properties.

- **Summable Property**: Firstly, we establish the summable property.

  From (2) in Algorithm 1, we have

  $$\|x_{i,k+1}(t) - x^*\|^2$$
  $$= \|x_{i,k}(t) - x^*\|^2 - 2\alpha \langle \nabla f_i(x_{i,k}(t)), x_{i,k}(t) - x^* \rangle + \alpha^2 \|\nabla f_i(x_{i,k}(t))\|^2. \tag{10}$$

  From the convexity of $f_i(x)$, we have

  $$-2\alpha \langle \nabla f_i(x_{i,k}(t)), x_{i,k}(t) - x^* \rangle \leq 2\alpha \{f_i(x^*) - f_i(x_{i,k}(t))\}. \tag{11}$$

  Using the strong growth condition (see Assumption 3) yields

  $$\|\nabla f_i(x^*)\| = 0 \tag{12}$$

  for any $i \in \mathcal{S}$ and $x^* \in \mathcal{X}^*$.

  Then, from Lemma 2 and (12), we have

  $$\|\nabla f_i(x_{i,k}(t))\|^2 \leq 2L_i \Big\{ f_i(x_{i,k}(t)) - f_i(x^*) \Big\}. \tag{13}$$

  Combining (10), (11), and (13), we arrive at

  $$\|x_{i,k+1}(t) - x^*\|^2 \leq \|x_{i,k}(t) - x^*\|^2 + (2\alpha - 2L_i\alpha^2)\Big\{ f_i(x^*) - f_i(x_{i,k}(t)) \Big\}. \tag{14}$$

  It is worth noting that the following inequality holds

  $$\|\bar{x}(t+1) - x^*\|^2 \leq \frac{1}{N} \sum_{i=1}^{N} \|x_{i,\tau}(t) - x^*\|^2.$$

  Thus, from Algorithm 1 and (14), we have

  $$\|\bar{x}(t+1) - x^*\|^2$$
  $$\leq \frac{1}{N} \sum_{i=1}^{N} (2\alpha - 2L_i\alpha^2) \sum_{k=0}^{\tau-1} \Big\{ f_i(x^*) - f_i(x_{i,k}(t)) \Big\} + \|\bar{x}(t) - x^*\|^2. \tag{15}$$

Under the stepsize setting (3), we have

$$2\alpha - 2L_i\alpha^2 > 0$$

for any $i \in \mathcal{S}$. Moreover, from Assumption 3, we have $\|\nabla f_i(x^*)\| = 0$ for any $x^* \in \mathcal{X}^*$ and $i \in \mathcal{S}$. Thus, in (15), we have

$$f_i(x^*) - f_i(x_{i,k}(t)) \leq 0$$

for any $i \in \mathcal{S}$ and $k = 0, 1, \cdots, \tau - 1$.

Thus, using (15), we have

$$\|\bar{x}(t+1) - x^*\|^2 \leq \|\bar{x}(t) - x^*\|^2 + \left\{2\alpha - 2L\alpha^2\right\}\left\{f(x^*) - f(\bar{x}(t))\right\}. \tag{16}$$

From (16), we can obtain

$$f(\bar{x}(t)) - f(x^*) \leq \frac{\|\bar{x}(t) - x^*\|^2 - \|\bar{x}(t+1) - x^*\|^2}{2\alpha - 2L_i\alpha^2}.$$

Thus, for any $T \geq 1$, we have

$$\sum_{t=1}^{T} \left\{f(\bar{x}(t)) - f(x^*)\right\} \leq \frac{\|\bar{x}(1) - x^*\|^2}{2\alpha - 2L_i\alpha^2}, \tag{17}$$

which establishes the summable property of the sequence $\{f(\bar{x}(t)) - f(x^*)\}$.

- **Monotonically Decreasing**:

  Next, we show that $f(\bar{x}(t))$ is monotonically decreasing.

  From Algorithm 3, we have

  $$\|x_{i,k+1}(t) - \bar{x}(t)\|$$
  $$\leq \|x_{i,k}(t) - \bar{x}(t) - \alpha\left(\nabla f_i(x_{i,k}(t)) - \nabla f_i(\bar{x}(t))\right)\| + \alpha\|\nabla f_i(\bar{x}(t))\|.$$

  Using Lemma 3 and the stepsize setting in (3), we arrive at

  $$\|x_{i,k+1}(t) - \bar{x}(t)\| \leq \|x_{i,k}(t) - \bar{x}(t)\| + \alpha\|\nabla f_i(\bar{x}(t))\|. \tag{18}$$

  Using the update rule in Algorithm 1, we obtain

  $$\|x_{i,k}(t) - \bar{x}(t)\| \leq k\alpha\|\nabla f_i(\bar{x}(t))\| \tag{19}$$

  for $k = 0, 1, 2, \cdots, \tau - 1$.

  It is worth noting that the following inequality always holds:

  $$\|\frac{1}{N}\sum_{i=1}^{N}\sum_{k=0}^{\tau-1}\nabla f_i(x_{i,k}(t))\| \leq \frac{1}{N}\sum_{i=1}^{N}\sum_{k=0}^{\tau-1}\left\|\nabla f_i(x_{i,k}(t)) - \nabla f_i(\bar{x}(t))\right\| + \tau\|\nabla f(\bar{x}(t))\|.$$

  From Assumption 1 and (3), we have

  $$\|\frac{1}{N}\sum_{i=1}^{N}\sum_{k=0}^{\tau-1}\nabla f_i(x_{i,k}(t))\| \leq \frac{1}{N}\sum_{i=1}^{N}\sum_{k=0}^{\tau-1}L_i\|x_{i,k}(t) - \bar{x}(t)\| + \tau\|\nabla f(\bar{x}(t))\|.$$

  Further using Assumption 3 and the update rull in Algorithm 3, we arrive at

  $$\|\frac{1}{N}\sum_{i=1}^{N}\sum_{k=0}^{\tau-1}\nabla f_i(x_{i,k}(t))\| \leq \left\{\tau + \frac{\eta(\tau-1)}{2}\right\}\|\nabla f(\bar{x}(t))\|. \tag{20}$$

  From (2), we have

  $$\bar{x}(t+1) = \bar{x}(t) - \frac{\alpha}{N}\sum_{i=1}^{N}\sum_{k=0}^{\tau-1}\nabla f_i(x_{i,k}(t)).$$

The global loss function $f(x)$ is $L$-smooth with $L = \frac{1}{N}\sum_{i=1}^{N} L_i$. From Lemma 4 and Assumption 1, we have

$$f(\bar{x}(t+1))$$

$$\leq f(\bar{x}(t)) - \left\langle \nabla f(\bar{x}(t)), \frac{1}{N}\sum_{i=1}^{N}\sum_{k=0}^{\tau-1}\alpha\nabla f_i(\bar{x}(t)) \right\rangle + \frac{L}{2}\left\| \frac{\alpha}{N}\sum_{i=1}^{N}\sum_{k=0}^{\tau-1}\nabla f_i(x_{i,k}(t)) \right\|^2$$

$$- \left\langle \nabla f(\bar{x}(t)), \frac{1}{N}\sum_{i=1}^{N}\sum_{k=0}^{\tau-1}\alpha\nabla f_i(x_{i,k}(t)) - \frac{1}{N}\sum_{i=1}^{N}\sum_{k=0}^{\tau-1}\alpha\nabla f_i(\bar{x}(t)) \right\rangle.$$

Further using Assumption 3 and (20), we have

$$f(\bar{x}(t+1))$$

$$\leq f(\bar{x}(t)) - \alpha\tau\|\nabla f(\bar{x}(t))\|^2 + \frac{L}{2}\alpha^2(t)\left\{ \tau + \frac{\eta(\tau-1)}{2} \right\}^2 \|\nabla f(\bar{x}(t))\|^2$$

$$+ L\alpha(t)\|\nabla f(\bar{x}(t))\|\left\{ \frac{1}{N}\sum_{i=1}^{N}\sum_{k=0}^{\tau-1}\|x_{i,k}(t) - \bar{x}(t)\| \right\}.$$

Based on the update rule in Algorithm 1, we arrive at

$$f(\bar{x}(t+1))$$

$$\leq f(\bar{x}(t)) - \alpha\tau\|\nabla f(\bar{x}(t))\|^2 + \frac{L}{2}\alpha^2\left\{ \tau + \frac{\eta(\tau-1)}{2} \right\}^2 \|\nabla f(\bar{x}(t))\|^2$$

$$+ L\alpha\|\nabla f(\bar{x}(t))\|\left\{ \eta\sum_{k=0}^{\tau-1}k\alpha\|\nabla f(\bar{x}(t))\| \right\},$$

which further implies

$$f(\bar{x}(t+1))$$

$$\leq f(\bar{x}(t)) - \alpha\tau\|\nabla f(\bar{x}(t))\|^2 + \frac{L}{2}\alpha^2\left\{ \tau + \frac{\eta(\tau-1)}{2} \right\}^2 \|\nabla f(\bar{x}(t))\|^2$$

$$+ \frac{\eta L\tau(\tau-1)\alpha^2}{2}\|\nabla f(\bar{x}(t))\|^2.$$

Therefore, the stepsize should satisfy

$$-\tau\alpha + \frac{L}{2}\left\{ \tau + \frac{\eta(\tau-1)}{2} \right\}^2\alpha^2 + \frac{\eta L\tau(\tau-1)}{2}\alpha^2 \leq 0$$

to guarantee the monotonically decreasing property of $f(\bar{x}(t)) - f(x^*)$. Equivalently, stepsize satisfying

$$\alpha \leq \frac{8\tau}{L(2\tau + \eta(\tau-1))^2 + 4\eta L\tau(\tau-1)}$$

guarantees the monotonically decreasing property of $f(\bar{x})$.

□

## D    PROOF OF THEOREM 2

*Proof.* From (2) in Algorithm 1, we can obtain

$$\|x_{i,k+1}(t) - x^*\|^2$$

$$= \|x_{i,k}(t) - x^*\|^2 - 2\alpha_i\langle\nabla f_i(x_{i,k}(t)), x_{i,k}(t) - x^*\rangle + \alpha_i^2\|\nabla f_i(x_{i,k}(t))\|^2. \tag{21}$$

From the convexity property of $f_i(x)$, we have

$$-2\alpha_i \langle \nabla f_i(x_{i,k}(t)), x_{i,k}(t) - x^* \rangle \le 2\alpha_i \{f_i(x^*) - f_i(x_{i,k}(t))\}. \tag{22}$$

Assumption 3 implies

$$\|\nabla f_i(x^*)\| = 0$$

for any $i \in \mathcal{S}$ and $x^* \in \mathcal{X}^*$. Thus, combining with Lemma 2, we have

$$\|\nabla f_i(x_{i,k}(t))\|^2 \le 2L_i \Big\{ f_i(x_{i,k}(t)) - f_i(x^*) \Big\}. \tag{23}$$

Combining (21), (22), and (23), we can obtain

$$\|x_{i,k+1}(t) - x^*\|^2 \le \|x_{i,k}(t) - x^*\|^2 + (2\alpha_i - 2L_i\alpha_i^2)\Big\{ f_i(x^*) - f_i(x_{i,k}(t)) \Big\}. \tag{24}$$

Note that the following inequality always holds:

$$\|\bar{x}(t+1) - x^*\|^2 \le \frac{1}{N} \sum_{i=1}^{N} \|x_{i,\tau}(t) - x^*\|^2.$$

Hence, Algorithm 1 and (24) imply

$$\|\bar{x}(t+1) - x^*\|^2$$
$$\le \frac{1}{N} \sum_{i=1}^{N} (2\alpha_i - 2L_i\alpha_i^2) \sum_{k=0}^{\tau-1} \Big\{ f_i(x^*) - f_i(x_{i,k}(t)) \Big\} + \|\bar{x}(t) - x^*\|^2. \tag{25}$$

Under the stepsize setting (4), we have

$$2\alpha_i - 2L_i\alpha_i^2 > 0$$

for any $i \in \mathcal{S}$. Moreover, Assumption 3 ensures

$$f_i(x^*) - f_i(x_{i,k}(t)) \le 0$$

for any $i \in \mathcal{S}$ and $k = 0, 1, \cdots, \tau - 1$.

Substituting the above inequality into (25) yields

$$\|\bar{x}(t+1) - x^*\|^2 \le \|\bar{x}(t) - x^*\|^2 + \min_{1 \le i \le N} \Big\{ 2\alpha_i - 2L_i\alpha_i^2 \Big\}\Big\{ f(x^*) - f(\bar{x}(t)) \Big\}. \tag{26}$$

From (26), we can obtain

$$f(\bar{x}(t)) - f(x^*) \le \frac{\|\bar{x}(t) - x^*\|^2 - \|\bar{x}(t+1) - x^*\|^2}{\min_{1 \le i \le N} \{2\alpha_i - 2L_i\alpha_i^2\}}.$$

Thus, for any $T \ge 1$, we have

$$\sum_{t=1}^{T} \Big\{ f(\bar{x}(t)) - f(x^*) \Big\} \le \frac{\|\bar{x}(1) - x^*\|^2}{\min_{1 \le i \le N} \{2\alpha_i - 2L_i\alpha_i^2\}}. \tag{27}$$

Since $f(\bar{x}(t)) - f(x^*) \ge 0$ holds for any $t$, we have

$$\lim_{t \to \infty} f(\bar{x}(t)) = f(x^*).$$

In addition, from (27), for any $T \ge 1$, we can obtain

$$f(\frac{1}{T}\sum_{t=1}^{T} \bar{x}(t)) - f(x^*) \le \frac{1}{T}\sum_{t=1}^{T} \Big\{ f(\bar{x}(t)) - f(x^*) \Big\} \le \frac{\|\bar{x}(1) - x^*\|^2}{\min_{1 \le i \le N} \{2\alpha_i - 2L_i\alpha_i^2\}T},$$

which completes the proof.

$\square$

# E   PROOF OF COROLLARY 3

*Proof.* We use $g_i(x)$ to represent the unbiased estimate of the gradient $\nabla f_i(x)$. From (2) in Algorithm 1, we can obtain

$$\|x_{i,k+1}(t) - x^*\|^2$$
$$\leq \|x_{i,k}(t) - x^*\|^2 - 2\alpha_i\langle\nabla f_i(x_{i,k}(t)), x_{i,k}(t) - x^*\rangle + 2\alpha_i^2\|\nabla f_i(x_{i,k}(t))\|^2$$
$$- 2\alpha_i\langle g_i(x_{i,k}(t)) - \nabla f_i(x_{i,k}(t)), x_{i,k}(t) - x^*\rangle + 2\alpha_i^2\|g_i(x_{i,k}(t)) - \nabla f_i(x_{i,k}(t))\|^2. \quad (28)$$

Using the convexity of $f_i(x)$, we arrive at

$$-2\alpha_i\langle\nabla f_i(x_{i,k}(t)), x_{i,k}(t) - x^*\rangle \leq 2\alpha_i\{f_i(x^*) - f_i(x_{i,k}(t))\}. \quad (29)$$

Assumption 3 implies

$$\|\nabla f_i(x^*)\| = 0$$

for any $i \in \mathcal{S}$ and $x^* \in \mathcal{X}^*$. Thus, combining the preceding relation with Lemma 2, we can obtain

$$\|\nabla f_i(x_{i,k}(t))\|^2 \leq 2L_i\{f_i(x_{i,k}(t)) - f_i(x^*)\}. \quad (30)$$

Combining (28), (29), and (30), we arrive at

$$\|x_{i,k+1}(t) - x^*\|^2$$
$$\leq \|x_{i,k}(t) - x^*\|^2 + (2\alpha_i - 2L_i\alpha_i^2)\{f_i(x^*) - f_i(x_{i,k}(t))\}$$
$$- 2\alpha_i\langle g_i(x_{i,k}(t)) - \nabla f_i(x_{i,k}(t)), x_{i,k}(t) - x^*\rangle + 2\alpha_i^2\|g_i(x_{i,k}(t)) - \nabla f_i(x_{i,k}(t))\|^2. \quad (31)$$

Using (31) and the property of the stochastic gradient, we have

$$\mathbb{E}[\|x_{i,k+1}(t) - x^*\|^2]$$
$$\leq \mathbb{E}[\|x_{i,k}(t) - x^*\|^2] + (2\alpha_i - 2L_i\alpha_i^2)\{f_i(x^*) - \mathbb{E}[f_i(x_{i,k}(t))]\} + 2\alpha_i^2\sigma^2. \quad (32)$$

Note that the following inequality always holds:

$$\mathbb{E}[\|\bar{x}(t+1) - x^*\|^2] \leq \frac{1}{N}\sum_{i=1}^N \mathbb{E}[\|x_{i,\tau}(t) - x^*\|^2].$$

Using the update rule in Algorithm 1 and (31), we arrive at

$$\mathbb{E}[\|\bar{x}(t+1) - x^*\|^2]$$
$$\leq \frac{1}{N}\sum_{i=1}^N(2\alpha_i - 2L_i\alpha_i^2)\sum_{k=0}^{\tau-1}\left\{f_i(x^*) - \mathbb{E}[f_i(x_{i,k}(t))]\right\} + \mathbb{E}[\|\bar{x}(t) - x^*\|^2] + \frac{1}{N}\sum_{i=1}^N 2\tau\alpha_i^2\sigma^2. \tag{33}$$

Under the stepsize setting (4), we can obtain $2\alpha_i - 2L_i\alpha_i^2 > 0$ for any $i \in \mathcal{S}$.

Moreover, Assumption 3 ensures $f_i(x^*) - \mathbb{E}[f_i(x_{i,k}(t))] \leq 0$ for any $i \in \mathcal{S}$ and $k = 0, 1, \cdots, \tau-1$.

Thus, from (33), we have

$$\mathbb{E}[\|\bar{x}(t+1) - x^*\|^2] \leq \min_{1\leq i\leq N}\left\{2\alpha_i - 2L_i\alpha_i^2\right\}\left\{f(x^*) - \mathbb{E}[f(\bar{x}(t))]\right\}$$
$$+ \frac{1}{N}\sum_{i=1}^N 2\tau\alpha_i^2\sigma^2 + \mathbb{E}[\|\bar{x}(t) - x^*\|^2], \quad (34)$$

which further implies

$$\mathbb{E}[f(\bar{x}(t))] - f(x^*) \leq \frac{\mathbb{E}[\|\bar{x}(t) - x^*\|^2] - \mathbb{E}[\|\bar{x}(t+1) - x^*\|^2]}{\min_{1\leq i\leq N}\{2\alpha_i - 2L_i\alpha_i^2\}} + \frac{\frac{1}{N}\sum_{i=1}^N 2\tau\alpha_i^2}{\min_{1\leq i\leq N}\{2\alpha_i - 2L_i\alpha_i^2\}}\sigma^2.$$

Thus, for any $T \geq 1$, we have

$$\sum_{t=1}^{T} \left\{ \mathbb{E}[f(\bar{x}(t))] - f(x^*) \right\} \leq \frac{\|\bar{x}(1) - x^*\|^2}{\min_{1 \leq i \leq N}\{2\alpha_i - 2L_i\alpha_i^2\}} + \frac{\frac{1}{N}\sum_{i=1}^{N} 2\tau\alpha_i^2 T}{\min_{1 \leq i \leq N}\{2\alpha_i - 2L_i\alpha_i^2\}}\sigma^2. \quad (35)$$

From (35), for any $T \geq 1$, we obtain

$$\mathbb{E}\left[f\left(\frac{1}{T}\sum_{t=1}^{T}\bar{x}(t)\right)\right] - f(x^*) \leq \frac{\|\bar{x}(1) - x^*\|^2}{\min_{1 \leq i \leq N}\{2\alpha_i - 2L_i\alpha_i^2\}T} + \frac{\frac{1}{N}\sum_{i=1}^{N} 2\tau\alpha_i^2}{\min_{1 \leq i \leq N}\{2\alpha_i - 2L_i\alpha_i^2\}}\sigma^2,$$

which completes the proof.

$\square$

## F  PROOF THEOREM 4

*Proof.* From Algorithm 2, the updated rules (6) and (7) can be equivalently expressed as

$$x_{i,k+1}(t) = x_{i,k}(t) - \alpha(\nabla f(\bar{x}(t)) - \nabla f_i(\bar{x}(t)) + \nabla f_i(x_{i,k}(t))). \quad (36)$$

The relation in (36), we further implies

$$\|x_{i,k+1}(t) - \bar{x}(t)\|$$
$$\leq \left\|x_{i,k}(t) - \bar{x}(t) + \alpha\Big(\nabla f_i(x_{i,k}(t)) - \nabla f_i(\bar{x}(t))\Big)\right\| + \alpha(t)\|\nabla f(\bar{x}(t))\|. \quad (37)$$

From Lemma 3 and (37), if the stepsize satisfies $0 \leq \alpha \leq \frac{1}{L_i}$, we have

$$\|x_{i,k+1}(t) - \bar{x}(t)\| \leq \|x_{i,k}(t) - \bar{x}(t)\| + \alpha\|\nabla f(\bar{x}(t))\|. \quad (38)$$

Using induction, we obtain

$$\|x_{i,k}(t) - \bar{x}(t)\| \leq k\alpha\|\nabla f(\bar{x}(t))\|. \quad (39)$$

Using the update rule in Algorithm 2, we can obtain

$$x_{i,\tau}(t) = \bar{x}(t) - \alpha\sum_{k=0}^{\tau-1}\nabla f_i(x_{i,k}(t)) - \tau\alpha\Big(\nabla f(\bar{x}(t)) - \nabla f_i(\bar{x}(t))\Big).$$

Therefore, the average parameter $\bar{x}(t+1)$ satisfies

$$\bar{x}(t+1) = \frac{1}{N}\sum_{i=1}^{N}x_{i,\tau}(t) = \bar{x}(t) - \frac{\alpha}{N}\sum_{i=1}^{N}\sum_{k=0}^{\tau-1}\nabla f_i(x_{i,k}(t)),$$

which further implies

$$\|\bar{x}(t+1) - x^*\|^2 - \|\bar{x}(t) - x^*\|^2$$
$$= -2\langle\frac{\alpha}{N}\sum_{i=1}^{N}\sum_{k=0}^{\tau-1}\nabla f_i(x_{i,k}(t)), \bar{x}(t) - x^*\rangle + \|\frac{\alpha}{N}\sum_{i=1}^{N}\sum_{k=0}^{\tau-1}\nabla f_i(x_{i,k}(t))\|^2. \quad (40)$$

For the first term on the right hand side of (40), we have

$$-\langle\frac{1}{N}\sum_{i=1}^{N}\sum_{k=0}^{\tau-1}\nabla f_i(x_{i,k}(t)), \bar{x}(t) - x^*\rangle$$
$$= \frac{1}{N}\sum_{i=1}^{N}\sum_{k=0}^{\tau-1}\langle x^* - x_{i,k}(t), \nabla f_i(x_{i,k}(t))\rangle + \frac{1}{N}\sum_{i=1}^{N}\sum_{k=0}^{\tau-1}\langle x_{i,k}(t) - \bar{x}(t), \nabla f_i(\bar{x}(t))\rangle$$
$$+ \frac{1}{N}\sum_{i=1}^{N}\sum_{k=0}^{\tau-1}\langle x_{i,k}(t) - \bar{x}(t), \nabla f_i(x_{i,k}(t)) - \nabla f_i(\bar{x}(t))\rangle. \quad (41)$$

Furthermore, using Assumption 1 and the convexity property of $f_i(x)$, we can obtain

$$-\langle \frac{1}{N} \sum_{i=1}^{N} \sum_{k=0}^{\tau-1} \nabla f_i(x_{i,k}(t)), \bar{x}(t) - x^* \rangle$$

$$\leq \frac{1}{N} \sum_{i=1}^{N} \sum_{k=0}^{\tau-1} \Big\{ f_i(x^*) - f_i(x_{i,k}(t)) \Big\} + \frac{1}{N} \sum_{i=1}^{N} \sum_{k=0}^{\tau-1} \Big\{ f_i(x_{i,k}(t)) - f_i(\bar{x}(t)) \Big\}$$

$$+ \frac{1}{N} \sum_{i=1}^{N} \sum_{k=0}^{\tau-1} L_i \|x_{i,k}(t) - \bar{x}(t)\|^2. \tag{42}$$

Combining (39) and (42), we arrive at

$$-\langle \frac{1}{N} \sum_{i=1}^{N} \sum_{k=0}^{\tau-1} \nabla f_i(x_{i,k}(t)), \bar{x}(t) - x^* \rangle$$

$$\leq \frac{1}{N} \sum_{i=1}^{N} \sum_{k=0}^{\tau-1} \Big\{ f_i(x^*) - f_i(\bar{x}(t)) \Big\} + \frac{1}{N} \sum_{i=1}^{N} \sum_{k=0}^{\tau-1} L_i k^2 \alpha^2 \|\nabla f(\bar{x}(t))\|^2. \tag{43}$$

Plugging the stepsize condition $0 < \alpha L_i \leq 1$ into (43) yields

$$-\langle \frac{1}{N} \sum_{i=1}^{N} \sum_{k=0}^{\tau-1} \nabla f_i(x_{i,k}(t)), \bar{x}(t) - x^* \rangle \leq \tau \Big\{ f(x^*) - f(\bar{x}(t)) \Big\} + \alpha \|\nabla f(\bar{x}(t))\|^2 \sum_{k=0}^{\tau-1} k^2. \tag{44}$$

Applying the relation $\sum_{k=1}^{n} k^2 = \frac{n(n+1)(2n+1)}{6}$ to the second term on the right hand side of (44) yields

$$-\langle \frac{1}{N} \sum_{i=1}^{N} \sum_{k=0}^{\tau-1} \nabla f_i(x_{i,k}(t)), \bar{x}(t) - x^* \rangle \leq \tau \Big\{ f(x^*) - f(\bar{x}(t)) \Big\} + A_1 \alpha \|\nabla f(\bar{x}(t))\|^2, \tag{45}$$

where $A_1 = \frac{\tau(\tau-1)(2\tau-1)}{6}$.

For the second term on the right hand side of (40), we have

$$\Big\| \frac{1}{N} \sum_{i=1}^{N} \sum_{k=0}^{\tau-1} \nabla f_i(x_{i,k}(t)) \Big\| \leq \frac{1}{N} \sum_{i=1}^{N} \sum_{k=0}^{\tau-1} \Big\| \nabla f_i(x_{i,k}(t)) - \nabla f_i(\bar{x}(t)) \Big\| + \tau \|\nabla f(\bar{x}(t))\|. \tag{46}$$

Using the smoothness condition in Assumption 1, we can further obtain

$$\Big\| \frac{1}{N} \sum_{i=1}^{N} \sum_{k=0}^{\tau-1} \nabla f_i(x_{i,k}(t)) \Big\| \leq \frac{1}{N} \sum_{i=1}^{N} \sum_{k=0}^{\tau-1} L_i \|x_{i,k}(t) - \bar{x}(t)\| + \tau \|\nabla f(\bar{x}(t))\|. \tag{47}$$

Combining (39) and (47), we can obtain

$$\Big\| \frac{1}{N} \sum_{i=1}^{N} \sum_{k=0}^{\tau-1} \nabla f_i(x_{i,k}(t)) \Big\| \leq \sum_{k=0}^{\tau-1} L_i k \alpha \|\nabla f(\bar{x}(t))\| + \tau \|\nabla f(\bar{x}(t))\|. \tag{48}$$

Applying the stepsize condition $0 < \tau \alpha L_i \leq 1$ to (48) yields

$$\Big\| \frac{1}{N} \sum_{i=1}^{N} \sum_{k=0}^{\tau-1} \nabla f_i(x_{i,k}(t)) \Big\| \leq A_2 \|\nabla f(\bar{x}(t))\|, \tag{49}$$

where $A_2 = 2\tau - 1$.

Then, combining (40), (45), and (49), we can obtain

$$\|\bar{x}(t+1) - x^*\|^2 - \|\bar{x}(t) - x^*\|^2$$

$$\leq 2\tau\alpha\Big\{f(x^*) - f(\bar{x}(t))\Big\} + \Big\{2A_1 + A_2^2\Big\}\alpha^2\|\nabla f(\bar{x}(t))\|^2. \tag{50}$$

Using the relation established in Lemma 5, we can obtain the following inequality from (50):

$$\|\bar{x}(t+1) - x^*\|^2 - \|\bar{x}(t) - x^*\|^2$$

$$\leq 2\tau\alpha\Big\{f(x^*) - f(\bar{x}(t))\Big\} + \frac{2A_1 + A_2^2}{\gamma}\Big\{f(\bar{x}(t)) - f(\bar{x}(t+1))\Big\}. \tag{51}$$

Rearranging terms yields

$$2\tau\alpha\Big\{f(\bar{x}(t)) - f(x^*)\Big\}$$

$$\leq \|\bar{x}(t) - x^*\|^2 - \|\bar{x}(t+1) - x^*\|^2 + \frac{2A_1 + A_2^2}{\gamma}\Big\{f(\bar{x}(t)) - f(\bar{x}(t+1))\Big\}. \tag{52}$$

Thus, for any $T > 0$, summarizing (52) from $t = 1$ to $t = T$ leads to

$$\sum_{t=1}^{T}\Big\{f(\bar{x}(t)) - f(x^*)\Big\} \leq \frac{1}{2\tau\alpha}\|\bar{x}(1) - x^*\|^2 + \frac{2A_1 + A_2^2}{2\tau\alpha\gamma}\Big\{f(\bar{x}(1)) - f(x^*)\Big\}. \tag{53}$$

Using (53), Lemma 5, and Lemma 1, we can conclude that $f(\bar{x}(t))$ converges to $f(x^*)$ with the convergence rate $o(1/t)$, which completes the proof. $\qquad\square$

# G  SUPPORTING LEMMAS FOR THE PROOF OF THEOREM 4

**Lemma 5.** *If the stepsize $\alpha$ of Algorithm 2 satisfies $0 < \alpha < \frac{2}{5L\tau - L}$, there exists a constant $\gamma > 0$ such that*

$$\gamma\alpha^2\|\nabla f(\bar{x}(t))\|^2 \leq f(\bar{x}(t)) - f(\bar{x}(t+1)).$$

*Moreover, the sequence $f(\bar{x}(t))$ is monotonically decreasing.*

*Proof.* Under Assumption 1, we know that $f(x)$ is $L$-smooth. Thus, from Lemma 4, we have

$$f(\bar{x}(t+1))$$

$$\leq f(\bar{x}(t)) - \alpha\langle\nabla f(\bar{x}(t)), \frac{1}{N}\sum_{i=1}^{N}\sum_{k=0}^{\tau-1}\nabla f_i(x_{i,k}(t))\rangle + \frac{L}{2}\|\frac{\alpha}{N}\sum_{i=1}^{N}\sum_{k=0}^{\tau-1}\nabla f_i(x_{i,k}(t))\|^2. \tag{54}$$

Substituting (49) into (54) leads to

$$f(\bar{x}(t+1))$$

$$\leq f(\bar{x}(t)) - \alpha\tau\|\nabla f(\bar{x}(t))\|^2 + 2L\tau^2\alpha^2\|\nabla f(\bar{x}(t))\|^2$$

$$+ \alpha\|\nabla f(\bar{x}(t))\|\Big\{\frac{1}{N}\sum_{i=1}^{N}\sum_{k=0}^{\tau-1}\|\nabla f_i(x_{i,k}(t)) - \nabla f_i(\bar{x}(t))\|\Big\}. \tag{55}$$

Using the smoothness condition in Assumption 1, we can have the following relationship for (55):

$$f(\bar{x}(t+1))$$

$$\leq f(\bar{x}(t)) - \alpha\tau\|\nabla f(\bar{x}(t))\|^2 + 2L\tau^2\alpha^2\|\nabla f(\bar{x}(t))\|^2$$

$$+ \alpha\|\nabla f(\bar{x}(t))\|\Big\{\frac{1}{N}\sum_{i=1}^{N}\sum_{k=0}^{\tau-1}L_i\|x_{i,k}(t) - \bar{x}(t)\|\Big\}. \tag{56}$$

Plugging (39) into (56) yields

$$f(\bar{x}(t+1)) \leq f(\bar{x}(t)) - \alpha\tau\|\nabla f(\bar{x}(t))\|^2 + \frac{5\tau^2 - \tau}{2}L\alpha^2\|\nabla f(\bar{x}(t))\|^2. \tag{57}$$

Rearranging like terms leads to

$$\left\{\alpha\tau - \frac{5L\tau^2 - L\tau}{2}\alpha^2\right\}\|\nabla f(\bar{x}(t))\|^2 \leq f(\bar{x}(t)) - f(\bar{x}(t+1)). \tag{58}$$

Since the stepsize $\alpha$ satisfies $0 < \alpha < \frac{2}{5L\tau - L}$, there exist $\gamma > 0$ such that

$$\alpha\tau - \frac{5\tau^2 - \tau}{2}L\alpha^2 \geq \gamma\alpha^2.$$

Thus, we have

$$\gamma\alpha^2\|\nabla f(\bar{x}(t))\|^2 \leq f(\bar{x}(t)) - f(\bar{x}(t+1)),$$

implying that the sequence $f(\bar{x}(t))$ is monotonically decreasing, which completes the proof. $\square$

## H   PROOF OF COROLLARY 5

We use $g_i(x)$ to represent the unbiased estimate of the gradient $\nabla f_i(x)$. We need the following Lemma 6 to prove Corollary 5.

**Lemma 6.** *If the stepsize satisfies* $0 < \alpha < \min_{\{1 \leq j \leq N\}}\{\frac{1}{L_j}\}$, *we have*

$$\mathbb{E}[\|x_{i,h}(t) - \bar{x}(t)\|^2] \leq 12\tau^2 L\alpha^2\mathbb{E}[f(\bar{x}(t)) - f(x^*)] + 27\tau\alpha^2\sigma^2$$

*for* $0 \leq h < \tau$ *and* $i \in \mathcal{S}$.

*Proof.* From Algorithm 2, we can obtain

$$x_{i,k+1}(t) = x_{i,k}(t) - \alpha\left\{\frac{1}{N}\sum_{j=1}^{N}g_j(\bar{x}(t)) - g_i(\bar{x}(t)) + g_i(x_{i,k}(t))\right\}.$$

Thus, we have

$$x_{i,k+1}(t) - \bar{x}(t)$$
$$= x_{i,k}(t) - \bar{x}(t) - \alpha\left\{\nabla f(\bar{x}(t)) - \nabla f_i(\bar{x}(t)) + \nabla f_i(x_{i,k}(t))\right\}$$
$$- \alpha\left\{\frac{1}{N}\sum_{j=1}^{N}g_j(\bar{x}(t)) - \frac{1}{N}\sum_{j=1}^{N}\nabla f_j(\bar{x}(t)) + \nabla f_i(\bar{x}(t)) - g_i(\bar{x}(t))\right.$$
$$\left. + g_i(x_{i,k}(t)) - \nabla f_i(x_{i,k}(t))\right\}.$$

From the property of stochastic gradients, we have

$$\mathbb{E}\left[\|x_{i,k+1}(t) - \bar{x}(t)\|^2\right]$$
$$= \mathbb{E}\left[\left\|x_{i,k}(t) - \bar{x}(t) - \alpha\left(\nabla f(\bar{x}(t)) - \nabla f_i(\bar{x}(t)) + \nabla f_i(x_{i,k}(t))\right)\right\|^2\right]$$
$$+ \alpha^2\mathbb{E}\left[\left\|\frac{1}{N}\sum_{j=1}^{N}g_j(\bar{x}(t)) - \frac{1}{N}\sum_{j=1}^{N}\nabla f_j(\bar{x}(t)) + \nabla f_i(\bar{x}(t)) - g_i(\bar{x}(t))\right.\right.$$
$$\left.\left. + g_j(x_{i,k}(t)) - \nabla f_i(x_{i,k}(t))\right\|^2\right]. \tag{59}$$

For the first term of the right hand side of (59), we can obtain

$$\mathbb{E}\left[\left\|x_{i,k}(t) - \bar{x}(t) - \alpha\left(\nabla f(\bar{x}(t)) - \nabla f_i(\bar{x}(t)) + \nabla f_i(x_{i,k}(t))\right)\right\|^2\right]$$

$$\leq (1 + \frac{1}{\tau})\mathbb{E}[\|x_{i,k}(t) - \bar{x}(t) - \alpha(\nabla f_i(x_{i,k}(t)) - \nabla f_i(\bar{x}(t)))\|^2] + (1+\tau)\alpha^2\mathbb{E}[\|\nabla f(\bar{x}(t))\|^2].$$

From Lemma 3, we can obtain

$$\mathbb{E}\left[\left\|x_{i,k}(t) - \bar{x}(t) - \alpha\left(\nabla f(\bar{x}(t)) - \nabla f_i(\bar{x}(t)) + \nabla f_i(x_{i,k}(t))\right)\right\|^2\right]$$

$$\leq (1 + \frac{1}{\tau})\mathbb{E}[\|x_{i,k}(t) - \bar{x}(t)\|^2] + (1+\tau)\alpha^2\mathbb{E}[\|\nabla f(\bar{x}(t))\|^2], \tag{60}$$

if the stepsize satisfies $0 < \alpha L_i \leq 1$ for any $1 \leq i \leq N$.

For the second term of the right hand side of (59), we have

$$\mathbb{E}\left[\left\|\frac{1}{N}\sum_{j=1}^{N} g_j(\bar{x}(t)) - \frac{1}{N}\sum_{j=1}^{N}\nabla f_j(\bar{x}(t)) + \nabla f_i(\bar{x}(t)) - g_i(\bar{x}(t))\right.\right.$$

$$\left.\left. + g_i(x_{i,k}(t)) - \nabla f_i(x_{i,k}(t))\right\|^2\right]$$

$$\leq \frac{3}{N}\sum_{j=1}^{N}\mathbb{E}[\|g_j(\bar{x}(t)) - \nabla f_j(\bar{x}(t))\|^2] + 3\mathbb{E}[\|\nabla f_i(\bar{x}(t)) - g_i(\bar{x}(t))\|^2]$$

$$+ 3\mathbb{E}[\|g_i(x_{i,k}(t)) - \nabla f_i(x_{i,k}(t))\|^2]$$

for any $i \in \mathcal{S}$.

Using the properties of stochastic gradients, we have

$$\mathbb{E}\left[\left\|\frac{1}{N}\sum_{j=1}^{N} g_j(\bar{x}(t)) - \frac{1}{N}\sum_{j=1}^{N}\nabla f_j(\bar{x}(t)) + \nabla f_i(\bar{x}(t)) - g_i(\bar{x}(t))\right.\right.$$

$$\left.\left. + g_i(x_{i,k}(t)) - \nabla f_i(x_{i,k}(t))\right\|^2\right] \leq 9\sigma^2 \tag{61}$$

for any $i \in \mathcal{S}$.

Combining (59), (60), and (61), we arrive at

$$\mathbb{E}[\|x_{i,k+1}(t) - \bar{x}(t)\|^2]$$

$$\leq (1 + \frac{1}{\tau})\mathbb{E}[\|x_{i,k}(t) - \bar{x}(t)\|^2] + (1+\tau)\alpha^2\mathbb{E}[\|\nabla f(\bar{x}(t))\|^2] + 9\alpha^2\sigma^2.$$

Using induction, we obtain the following relation holding for any $0 \leq k < \tau$:

$$\mathbb{E}\left[\|x_{i,k}(t) - \bar{x}(t)\|^2\right] \leq \left\{(1+\tau)\alpha^2\mathbb{E}[\|\nabla f(\bar{x}(t))\|^2] + 9\alpha^2\sigma^2\right\}\sum_{h=0}^{\tau-1}(1 + \frac{1}{\tau})^h,$$

which further implies

$$\mathbb{E}\left[\|x_{i,k}(t) - \bar{x}(t)\|^2\right] \leq \left\{(1+\tau)\alpha^2\mathbb{E}[\|\nabla f(\bar{x}(t))\|^2] + 9\alpha^2\sigma^2\right\}\frac{(1+\frac{1}{\tau})^\tau - 1}{(1+\frac{1}{\tau}) - 1}.$$

Using the relation $\|\nabla f(\bar{x}(t))\|^2 \leq 2L(f(\bar{x}(t)) - f(x^*))$ from Assumption 1 and the convex property of $f_i(x)$, we can obtain

$$\mathbb{E}\left[\|x_{i,k}(t) - \bar{x}(t)\|^2\right] \leq 12\tau^2 L\alpha^2\mathbb{E}[f(\bar{x}(t)) - f(x^*)] + 27\tau\alpha^2\sigma^2,$$

which completes the proof. $\qquad\square$

Next we proceed to prove Corollary 5. From the update rule in Algorithm 2, we have

$$\bar{x}(t+1) = \bar{x}(t) - \frac{\alpha}{N} \sum_{j=1}^{N} \sum_{h=0}^{\tau-1} g_j(x_{j,h}(t)),$$

which further implies

$$\|\bar{x}(t+1) - x^*\|^2 - \|\bar{x}(t) - x^*\|^2$$

$$= -2\alpha \langle \frac{1}{N} \sum_{j=1}^{N} \sum_{h=0}^{\tau-1} g_j(x_{j,h}(t)), x_i(k\tau) - x^* \rangle + \alpha^2 \| \frac{1}{N} \sum_{j=1}^{N} \sum_{h=0}^{\tau-1} g_j(x_{j,h}(t)) \|^2. \quad (62)$$

For the term $-2\alpha \langle \frac{1}{N} \sum_{j=1}^{N} \sum_{h=0}^{\tau-1} g_j(x_{j,h}(t)), x_i(k\tau) - x^* \rangle$ in (62), we have

$$-2\alpha \mathbb{E} \Big[ \langle \frac{1}{N} \sum_{j=1}^{N} \sum_{h=0}^{\tau-1} g_j(x_{j,h}(t)), x_i(k\tau) - x^* \rangle \Big]$$

$$= \frac{2\alpha}{N} \sum_{j=1}^{N} \sum_{h=0}^{\tau-1} \mathbb{E} \Big[ \langle x^* - x_{j,h}(t), \nabla f_j(x_{j,h}(t)) \rangle \Big] + \frac{2\alpha}{N} \sum_{j=1}^{N} \sum_{h=0}^{\tau-1} \mathbb{E} \Big[ \langle x_{j,h}(t) - \bar{x}(t), \nabla f_j(x_{j,h}(t)) \rangle \Big].$$

Using the convexity of $f_i(x)$ and Assumption 1, we arrive at

$$-2\alpha \mathbb{E} \Big[ \langle \frac{1}{N} \sum_{j=1}^{N} \sum_{h=0}^{\tau-1} g_j(x_{j,h}(t)), x_i(k\tau) - x^* \rangle \Big]$$

$$\leq \frac{2\alpha}{N} \sum_{j=1}^{N} \sum_{h=0}^{\tau-1} \mathbb{E} \Big[ f_j(x^*) - f_j(x_{j,h}(t)) \Big]$$

$$+ \frac{2\alpha}{N} \sum_{j=1}^{N} \sum_{h=0}^{\tau-1} \mathbb{E} \Big[ f_j(x_{j,h}(t)) - f_j(\bar{x}(t)) + \frac{L}{2} \|x_{j,h}(t) - \bar{x}(t)\|^2 \Big],$$

which further implies

$$-2\alpha \mathbb{E} \Big[ \langle \frac{1}{N} \sum_{j=1}^{N} \sum_{h=0}^{\tau-1} g_j(x_{j,h}(t)), x_i(k\tau) - x^* \rangle \Big]$$

$$\leq 2\alpha\tau \mathbb{E} \Big[ f(x^*) - f(\bar{x}(t)) \Big] + \frac{\alpha L}{N} \sum_{j=1}^{N} \sum_{h=0}^{\tau-1} \mathbb{E} \Big[ \|x_{j,h}(t) - \bar{x}(t)\|^2 \Big]. \quad (63)$$

From Lemma 6, we have

$$\mathbb{E}[\|x_{j,h}(t) - \bar{x}(t)\|^2] \leq 12\tau^2 L \alpha^2 \mathbb{E}[f(\bar{x}(t)) - f(x^*)] + 27\tau \alpha^2 \sigma^2 \quad (64)$$

for $1 \leq h < \tau$.

Combining (63) and (64), yields

$$-2\alpha \mathbb{E} \Big[ \langle \frac{1}{N} \sum_{j=1}^{N} \sum_{h=0}^{\tau-1} g_j(x_{j,h}(t)), \bar{x}(t) - x^* \rangle \Big]$$

$$\leq 2\alpha\tau \mathbb{E} \Big[ f(x^*) - f(\bar{x}(t)) \Big] + 12\tau^3 L^2 \alpha^3 \mathbb{E}[f(\bar{x}(t)) - f(x^*)] + 27\tau^2 L \alpha^3 \sigma^2.$$

When the stepsize satisfies $0 < 6\tau\alpha L \leq 1$, we have

$$-2\alpha \mathbb{E} \Big[ \langle \frac{1}{N} \sum_{j=1}^{N} \sum_{h=0}^{\tau-1} g_j(x_{j,h}(t)), \bar{x}(t) - x^* \rangle \Big]$$

$$\leq 2\alpha\tau \mathbb{E} \Big[ f(x^*) - f(\bar{x}(t)) \Big] + 2\tau^2 L \alpha^2 \mathbb{E}[f(\bar{x}(t)) - f(x^*)] + 9\tau^2 \alpha^2 \sigma^2. \quad (65)$$

For the term $\alpha^2 \|\frac{1}{N} \sum_{j=1}^{N} \sum_{h=0}^{\tau-1} g_j(x_{j,h}(t))\|^2$ in (62), we have

$$\alpha^2 \left\| \frac{1}{N} \sum_{j=1}^{N} \sum_{h=0}^{\tau-1} g_j(x_{j,h}(t)) \right\|^2$$

$$\leq 2\alpha^2 \left\| \frac{1}{N} \sum_{j=1}^{N} \sum_{h=0}^{\tau-1} \left\{ g_j(x_{j,h}(t)) - g_j(\bar{x}(t)) \right\} \right\|^2 + 2\alpha^2 \left\| \frac{1}{N} \sum_{j=1}^{N} \sum_{h=0}^{\tau-1} g_j(\bar{x}(t)) \right\|^2. \tag{66}$$

Using the smoothness conditon in Assumption 1 and the inequality $\|\sum_{i=1}^{k} a_i\|^2 \leq k \sum_{i=1}^{k} \|a_i\|^2$, we have

$$\alpha^2 \left\| \frac{1}{N} \sum_{j=1}^{N} \sum_{h=0}^{\tau-1} \left\{ g_j(x_{j,h}(t)) - g_j(\bar{x}(t)) \right\} \right\|^2$$

$$\leq \frac{3\tau L^2 \alpha^2}{N} \sum_{j=1}^{N} \sum_{h=0}^{\tau-1} \left\| x_{j,h}(t) - \bar{x}(t) \right\|^2 + \frac{3\tau \alpha^2}{N} \sum_{j=1}^{N} \sum_{h=0}^{\tau-1} \left\| g_j(x_{j,h}(t)) - \nabla f_j(x_{j,h}(t)) \right\|^2$$

$$+ \frac{3\tau \alpha^2}{N} \sum_{j=1}^{N} \sum_{h=0}^{\tau-1} \left\| \nabla f_j(\bar{x}(t)) - g_j(\bar{x}(t)) \right\|^2. \tag{67}$$

From (67) and the property of stochastic gradient, we have

$$2\alpha^2 \mathbb{E}\left[ \left\| \frac{1}{N} \sum_{j=1}^{N} \sum_{h=0}^{\tau-1} \left\{ g_j(x_{j,h}(t)) - g_j(\bar{x}(t)) \right\} \right\|^2 \right]$$

$$\leq \frac{6\tau \alpha^2 L^2}{N} \sum_{j=1}^{N} \sum_{h=0}^{\tau-1} \mathbb{E}\left[ \left\| x_{j,h}(t) - \bar{x}(t) \right\|^2 \right] + 12\alpha^2 \tau^2 \sigma^2. \tag{68}$$

Plugging the inequality in Lemma 6 into (68) leads to

$$2\alpha^2 \mathbb{E}\left[ \left\| \frac{1}{N} \sum_{j=1}^{N} \sum_{h=0}^{\tau-1} \left\{ g_j(x_{j,h}(t)) - g_j(\bar{x}(t)) \right\} \right\|^2 \right]$$

$$\leq 72\tau^4 L^3 \alpha^4 \mathbb{E}[f(\bar{x}(t)) - f(x^*)] + 162\tau^3 L^2 \alpha^4 \sigma^2 + 12\alpha^2 \tau^2 \sigma^2. \tag{69}$$

For the term $\|\frac{1}{N} \sum_{j=1}^{N} \sum_{h=0}^{\tau-1} g_j(x_j(k\tau))\|^2$ in (66), we have

$$2\alpha^2 \left\| \frac{1}{N} \sum_{j=1}^{N} \sum_{h=0}^{\tau-1} g_j(\bar{x}(t)) \right\|^2$$

$$\leq \frac{4\alpha^2 \tau^2}{N} \sum_{j=1}^{N} \|g_j(\bar{x}(t)) - \nabla f_j(\bar{x}(t))\|^2 + 4\alpha^2 \tau^2 \|\nabla f(\bar{x}(t))\|^2.$$

Using Lemma 6 and the property of stochastic gradients, we have

$$2\alpha^2 \mathbb{E}[\|\frac{1}{N} \sum_{j=1}^{N} \sum_{h=0}^{\tau-1} g_j(\bar{x}(t))\|^2] \leq 4\alpha^2 \tau^2 \sigma^2 + 8\alpha^2 \tau^2 L \mathbb{E}[f(\bar{x}(t)) - f(x^*)]. \tag{70}$$

Combining (66), (69), and (70), we have

$$\alpha^2 \mathbb{E}\left[ \left\| \frac{1}{N} \sum_{j=1}^{N} \sum_{h=0}^{\tau-1} g_j(x_{j,h}(t)) \right\|^2 \right]$$

$$\leq (72\tau^4 L^3 \alpha^4 + 8\alpha^2 \tau^2 L) \mathbb{E}[f(\bar{x}(t)) - f(x^*)] + 162\tau^3 L^2 \alpha^4 \sigma^2 + 16\alpha^2 \tau^2 \sigma^2.$$

When the stepsize satisfies $0 < 6\tau\alpha L \le 1$, we have

$$\alpha^2 \mathbb{E}\left[\left\|\frac{1}{N}\sum_{j=1}^{N}\sum_{h=0}^{\tau-1} g_j(x_{j,h}(t))\right\|^2\right] \le 10\tau^2 L\alpha^2 \mathbb{E}[f(\bar{x}(t)) - f(x^*)] + 25\alpha^2\tau^2\sigma^2. \qquad (71)$$

Combining (62), (65), and (71), we have

$$\mathbb{E}[\|\bar{x}(t+1) - x^*\|^2] - \mathbb{E}[\|\bar{x}(t) - x^*\|^2] \le (2\alpha\tau - 12\tau^2 L\alpha^2)\mathbb{E}[f(x^*) - f(\bar{x}(t))] + 34\tau^2\alpha^2\sigma^2. \qquad (72)$$

When the stepsize satisfies $0 < \alpha \le \frac{1}{12\tau L}$, we have $\alpha\tau - 12\tau^2 L\alpha^2 \ge 0$. Plugging the preceding inequality into (72) yileds

$$\frac{1}{T}\sum_{t=1}^{T} \mathbb{E}\left[f(\bar{x}(t)) - f(x^*)\right] \le \frac{\|x_i(1) - x^*\|^2}{\alpha\tau T} + \frac{34\tau^2\alpha^2}{\alpha\tau}\sigma^2.$$

Moreover, using the stepsize condition $0 < \alpha \le \min_{1 \le j \le N}\{\frac{1}{L_j}, \frac{1}{12\tau L}\}$ and the convexity of $f(x)$, we can obtain

$$\mathbb{E}\left[f\left(\frac{1}{T}\sum_{t=1}^{T}\bar{x}(t)\right)\right] - f(x^*) \le \frac{\|x(1) - x^*\|^2}{\alpha\tau T} + 34\tau\alpha\sigma^2$$

for any $i \in \mathcal{S}$, which completes the proof.

