# OpenReview forum: "Achieving Faster than $O(1/t)$ Convergence in Convex Federated Learning"
_ICLR.cc/2026/Conference — ICLR 2026 Conference Withdrawn Submission_

### Official Review · Reviewer_xGeQ · 2025-10-28

**Soundness:** 3
**Presentation:** 3
**Contribution:** 3
**Rating:** 2
**Confidence:** 4

**Summary:**

This paper provides a new framework for analyzing the convergence of federated learning under convex function. For non-IID data scenario, it proposes an algorithm for drift mitigation, where the algorithm is proven to achieve a convergence rate of o(1/t).

**Strengths:**

This work proposes a new framework for analyzing federated learning convergence under convex setting. The analytical framework shows a better bound, and can be extended to analyze the convergence of other algorithms.

**Weaknesses:**

- The analytical framework is the major contribution of this work. Thus, it would be important to detailed explain and justify the analytical framework in the main context, emphasizing on its main idea, novelty, and significance.
- For the analysis, a new assumption on Strong Growth Condition is involved. Although authors have justified this point using existing works, it is still important to show this assumption can hold in practical systems, i.e., actual training. Otherwise, proving a better bound with an unrealistic assumption is not meaningful. This is very important.
- Algorithm 2 does not seem to be reasonable. Although authors stated it as mitigating drift, it does not seem to be the case. Based on equations (6) and (7), x_{k+1} = x_k - \alpha(\nable f(x_{k}) - \nable f(x_{k-2})). This equation is obtained by substituting (7) into (6) in the next local update. This equation shows that the update is towards the direction of the gradient difference between x_k and x_{k-2}. This does not correspond to drift mitigation, and meanwhile, this direction does not seem to be the correct direction for update, which can be seen by drawing the update direction of a simple two-dimensional convex function.
- The baselines are mostly from 2020-2022, which are out-of-date.

Minor comment:
- In Remark 1, should the k's be t's?
- On page 5, SCAFFOLD rather than SCADDOLD.

**Questions:**

- Please emphasize/clarify on why the proposed analytical framework is better and how it improves the bound.
- Please justify the assumption on Strong Growth Condition through real-world experiments or evidence.
- Please explain Algorithm 2 (see the above comment).

---

### Official Review · Reviewer_ANjd · 2025-11-01

**Soundness:** 3
**Presentation:** 1
**Contribution:** 2
**Rating:** 2
**Confidence:** 4

**Summary:**

The paper provides a refined convergence analysis of convex federated learning algorithms, focusing on variants Federated Averaging (FedAvg) with client-drift compensation. The authors show improved dependence on the number of optimization steps -- from $O(1/t)$ to $o(1/t)$ under certain assumptions -- both in iid and non-iid data distribution and support their claims with the empirical evaluation.

**Strengths:**

The focus of the work -- the convergence analysis of the federated learning algorithms in the convex setup -- is relevant.

The paper attempts to address both IID and non-IID settings within a unified framework.

**Weaknesses:**

**The IID case:** analysis relies on a strong growth condition (Assumption 3) that is considerably stronger than standard assumptions in FL (e.g., bounded gradient variance or Hessian similarity). This assumption effectively implies an overparameterized regime where client drift is negligible, limiting the practical relevance of the claimed improvements.

The step-size gain ($4\times$ larger than [1]) appears to trade off against the stronger assumption, reducing the significance of the improvement.

**The non-IID case:** Algorithm 2 lacks clear algorithmic novelty, and the convergence rate (Corollary 5) does not capture the benefit of increasing the number of clients N, unlike prior work such as [1]. Moreover, the resulting convergence neighborhood appears larger than that of [1].

**Experiments:**
The empirical section adds limited insight, as the conservative nature of theoretical step sizes in FedAvg is already well known.

**Presentation:**
Presentation quality needs improvement: the paper has numerous typographical errors, inconsistent formalism, and an unpolished bibliography (e.g., just in line 53 there are 2 mistakes in the author names).

**Questions:**

- Could authors clarify whether the Assumption 3 can be relaxed?
- What is the concrete algorithmic novelty between your Algorithm 2 and other client-drift compensated SGD variants?
- How does your analysis compare quantitatively to [1] in non-iid setup?

---

### Official Review · Reviewer_Enro · 2025-11-01

**Soundness:** 2
**Presentation:** 3
**Contribution:** 2
**Rating:** 2
**Confidence:** 4

**Summary:**

This paper provides convergence analysis for convex federated learning. Convergence rates for both IID and non-IID scenarios are provided. Some experiments were provided to show convergence behavior.

**Strengths:**

- Developing tighter bounds for FL is an important and interesting topic to the theoretical ML community.

- The paper is well written and easy to follow.

**Weaknesses:**

There are major inaccurate statements regarding the theoretical contributions:

- At first sight, the reviewer was quite impressed by the authors' statement that they were able to achieve tighter bounds, even if Glasgow et al. 2022 provided a lower bound saying that FedAvg cannot achieve faster than $O(1/T)$.  Then the reviewer looked into the details of the proofs and realized that this paper focuses on batch GD instead of SGD, where the latter is more suitable in most scenarios such as cross-device FL. Importantly, the current results between the two schemes must be discussed separately, yet the authors confuse the both.

- In the first contribution bullet as well as lines 209-211, the authors stated that they used much larger step sizes compared to Qin et all 2022 and Khaled et al 2020. This is not a valid/fair comparison to the reviewer, as the paper assumes batch GD but the two references use SGD. The same issue applies to the third contribution bullet as well as the entire table 1.

- Although the authors provided convergence analysis for SGD (e.g., Corollary 3 and 5), the results included a non-decreasing term which yields a much looser bound than existing rates for FL with SGD.

Since this paper is largely theoretical with many inaccurate statements, I am afraid this current version is far from the publishable standard of this conference.

I also have some major concerns regarding the experiments:

- What do mean by Algorithm 2 in these figures? In lines 259-268, Algorithm 2 was termed as a general framework that models some existing approaches, but in Fig. 2-3, it is referred to as a specific method. Can the authors specify this detail?

- Fig. 2 and Fig. 4 look problematic. It would be surprising to see that all these algorithms achieve the same or even higher accuracy with a much higher level of data heterogeneity $\alpha=0.1$.

- In line 411, why is there an extremely small term $10^{-15}$ in the learning rate?

**Questions:**

See above.

**Details Of Ethics Concerns:**

I am reviewing two submissions (the other submission's ID is 4644) that appear to come from the same author group, based on writing style and content. Both papers make similar novelty claims, where each states that they are the first to provide a $o(1/t)$ convergence rates for FL with convex objectives. Since the two papers appear to overlap conceptually, I want to flag this in case it raises a concern about overlapping contributions. I am not making a judgment, but am thinking the chairs might want to review this.

---

### Note · Authors · 2025-11-16

I have read and agree with the venue's withdrawal policy on behalf of myself and my co-authors.